# Federated Conditional Stochastic Optimization

**Xidong Wu**[†]
Department of ECE
University of Pittsburgh
Pittsburgh, PA 15213
`xidong_wu@outlook.com`

**Jianhui Sun**[†]
Computer Science
University of Virginia
Charlottesville, VA 22903
`js9gu@virginia.edu`

**Zhengmian Hu**
Computer Science
University of Maryland
College Park, MD 20742
`huzhengmian@gmail.com`

**Junyi Li**
Department of ECE
University of Pittsburgh
Pittsburgh, PA 15213
`junyili.ai@gmail.com`

**Aidong Zhang**[§]
Computer Science
University of Virginia
Charlottesville, VA 22903
`aidong@virginia.edu`

**Heng Huang**[*]
Computer Science
University of Maryland
College Park, MD 20742
`henghuanghh@gmail.com`

## Abstract

Conditional stochastic optimization has found applications in a wide range of machine learning tasks, such as invariant learning, AUPRC maximization, and meta-learning. As the demand for training models with large-scale distributed data grows in these applications, there is an increasing need for communication-efficient distributed optimization algorithms, such as federated learning algorithms. This paper considers the nonconvex conditional stochastic optimization in federated learning and proposes the first federated conditional stochastic optimization algorithm (FCSG) with a conditional stochastic gradient estimator and a momentum-based algorithm (*i.e.*, FCSG-M). To match the lower bound complexity in the single-machine setting, we design an accelerated algorithm (Acc-FCSG-M) via the variance reduction to achieve the best sample and communication complexity. Compared with the existing optimization analysis for Meta-Learning in FL, federated conditional stochastic optimization considers the sample of tasks. Extensive experimental results on various tasks validate the efficiency of these algorithms.

## 1 Introduction

The conditional stochastic optimization arises throughout a wide range of machine learning tasks, such as the policy evaluation in reinforcement learning [5], invariant learning [16], instrumental variable regression in causal inference [30], Model-Agnostic Meta-Learning (MAML) [10], AUPRC maximization [28] and so on. Recently many efficient conditional stochastic optimization algorithms have been developed [16–18, 28, 34, 32] to solve the corresponding machine learning problems and applications. However, all existing conditional stochastic optimization algorithms were only designed for centralized learning (*i.e.*, model and data both deployed at a single machine) or finite-sum optimization, without considering the large-scale online distributed scenario. Many federated learning algorithms [26, 22, 29, 23, 44, 39, 25, 45] were proposed since FL is a communication-efficient training paradigm for large-scale machine learning training preserving data privacy. In federated learning, clients update the model locally, and the global server aggregates the model

---

[†]Equal contribution

[§]This work was partially supported by NSF CNS 2213700 and CCF 2217071 at UVA.

[*]This work was partially supported by NSF IIS 1838627, 1837956, 1956002, 2211492, CNS 2213701, CCF 2217003, DBI 2225775 at Pitt and UMD.

parameters periodically. Although federated learning has been actively applied to numerous real-world applications in the past years, the federated conditional stochastic optimization problem is still underexplored. To bridge this gap, in this paper we study the following federated conditional stochastic optimization problem:

$$\min_{x \in \mathcal{X}} F(x) := \frac{1}{N} \sum_{n=1}^{N} \mathbb{E}_{\xi^n} f_{\xi^n}^n (\mathbb{E}_{\eta^n | \xi^n} g_{\eta^n}^n (x, \xi^n)), \tag{1}$$

where $\mathcal{X} \subseteq \mathbb{R}^d$ is a closed convex set, $\mathbb{E}_{\xi^n} f_{\xi^n}^n (\cdot) : \mathbb{R}^{d'} \to \mathbb{R}$ is the outer-layer function on the $n$-th device with the randomness $\xi^n$, and $\mathbb{E}_{\eta^n | \xi^n} g_{\eta^n}^n (\cdot, \xi^n) : \mathbb{R}^d \to \mathbb{R}^{d'}$ is the inner-layer function on the $n$-th device with respect to the conditional distribution of $\eta^n \mid \xi^n$. We assume $f_\xi^n(\cdot)$ and $g_\eta^n(\cdot, \xi)$ are continuously differentiable. The objective subsumes two stochastic functions in (1), where the inner functions rely on the randomnesses of both inner and outer layers, and $\xi$ and $\eta$ are not independent, which makes the federated conditional stochastic optimization more challenging compared with the standard federated learning optimization problems.

Federated conditional stochastic optimization contains the standard federated learning optimization as a special situation when the inner-layer function $g_{\eta^n}^n(x, \xi^n) = x$. In addition, federated stochastic compositional optimization is similar to federated conditional stochastic optimization given that both problems contain two-layer nested expectations. However, they are fundamentally different. In federated stochastic compositional optimization, the inner randomness $\eta$ and the outer randomness $\xi$ are independent and data samples of the inner layer are available directly from $\eta$ (instead of a conditional distribution as in Problem (1)). Therefore, when randomnesses $\eta$ and $\xi$ are independent and $g_{\eta^n}^n(x, \cdot) = g_{\eta^n}^n(x)$, (1) is converted into federated stochastic compositional optimization [11].

Recently, to solve the conditional stochastic optimization problem efficiently, [16] studied the sample complexity of the sample average approximation for conditional stochastic optimization. Afterward, [17] proposed the algorithm called biased stochastic gradient descent (BSGD) and an accelerated algorithm called biased SpiderBoost (BSpiderBoost). The convergence guarantees of BSGD and BSpiderBoost under different assumptions are established. More recently, [28, 34, 37, 32, 14] reformulated the AUC maximization into a finite-sum version of conditional stochastic optimization and introduced algorithms to solve it. In an increasing amount of distributed computing settings, efficient federated learning algorithms are absolutely necessary but still lacking. An important example of conditional stochastic optimization is MAML. In meta-learning, we attempt to train models that can efficiently adapt to unseen tasks via learning with metadata from similar tasks [10]. When the tasks are distributed at different clients, a federated version of MAML would be beneficial to leverage information from all workers [3]. A lot of existing efforts [19, 11] have been made to convert FL MAML into federated compositional optimization. Nonetheless, they ignore the sample of tasks in MAML, and federated conditional stochastic optimization problems have never been studied. Thus, there exists a natural question: Can we design federated algorithms for conditional stochastic optimization while maintaining the fast convergence rate to solve Problem (1)?

In this paper, we give an affirmative answer to the above question. We propose a suite of approaches to solve Problem (1) and establish their corresponding convergence guarantee. To our best knowledge, this is the first work that thoroughly studies federated conditional stochastic optimization problems and provides completed theoretical analysis. Our proposed algorithm matches the lower-bound sample complexity in a single-machine setting and obtains convincing results in empirical experiments. Our main contributions are four-fold:

1) we propose the federated conditional stochastic gradient (FCSG) algorithm to solve Problem (1). We establish the theoretical convergence analysis for FCSG. In the general nonconvex setting, we prove that FCSG has a sample complexity of $O(\epsilon^{-6})$ and communication complexity of $O(\epsilon^{-3})$ to reach an $\epsilon$-stationary point, and achieves an appealing linear speedup *w.r.t* the number of clients.

2) To further improve the empirical performance of our algorithm, we introduce a momentum-based FCSG algorithm, called FCSG-M since the momentum-based estimator could reduce noise from samples with history information. FCSG-M algorithm obtains the same theoretical guarantees as FCSG.

3) To reach the lower bound of sample complexity of the single-machine counterpart [17], we propose an accelerated version of FCSG-M (Acc-FCSG-M) based on the momentum-based variance reduction technique. We prove that Acc-FCSG-M has a sample complexity

of $O(\epsilon^{-5})$, and communication complexity of $O(\epsilon^{-2})$, which matches the best sample complexity attained by single-machine algorithm BSpiderBoost with variance reduction.

4) Experimental results on the robust logistic regression, MAML and AUPRC maximization tasks validate the effectiveness of our proposed algorithms.

## 2 Related Work

### 2.1 Conditional Stochastic Optimization

[16] studied the generalization error bound and sample complexity of the sample average approximation (SAA) for conditional stochastic optimization. Subsequently, [17] proposed a class of efficient stochastic gradient-based methods for general conditional stochastic optimization to reach either a global optimal point in the convex setting or a stationary point in the nonconvex setting, respectively. In the nonconvex setting, BSGD has the sample complexity of $O(\epsilon^{-6})$ and a variance reduction algorithm (BSpiderBoost) has the sample complexity of $O(\epsilon^{-5})$. [18] utilized the Monte Carlo method to achieve better results compared with the vanilla stochastic gradient method. Recently, [28] converted AUPRC maximization optimization into the finite-sum version of the conditional stochastic optimization and propose adaptive and non-adaptive stochastic algorithms to solve it. Similarly, recent work [34] used moving average techniques to improve the convergence rate of AUPRC maximization optimization and provide theoretical analysis for the adaptive algorithm. Furthermore, [32] focused on finite-sum coupled compositional stochastic optimization, which limits the outer-layer function to the finite-sum structure. The algorithms proposed in [32] improved oracle complexity with the parallel speed-up. More recently, [14] use federated learning to solve AUC maximization. However, algorithms proposed in [28, 34, 32, 14] for AUC maximization have a significant limitation because they maintain an inner state for each data point. As a result, its convergence rate depends on the number of data points and cannot be extended to other tasks and large-scale model training. It is also not applicable to online learning due to the dependence on each local data point. [38] consider the decentralised online AUPRC maximization but the theoretical analysis cannot be applied into the federated learning.

### 2.2 Stochastic Compositional Optimization

Recently, a related optimization problem, stochastic compositional optimization, has attracted widely attention [36, 41, 11] and solve the following objective:
$$\min_{x \in \mathcal{X}} F(x) := \mathbb{E}_\xi f_\xi(\mathbb{E}_\eta g_\eta(x)). \tag{2}$$
To address this problem, [36] developed SCGD, which utilizes the moving average technique to estimate the inner-layer function value. [35] further developed an accelerated SCGD method with the extrapolation-smoothing scheme. Subsequently, a series of algorithms [20, 43, 15, 41] were presented to improve the complexities using the acceleration or variance reduction techniques.

More recently, [19] and [11] studied the stochastic compositional problem in federated learning. [19] transformed the distributionally robust federated learning problem (*i.e.*, a minimax optimization problem) into a simple compositional optimization problem by using KL divergence regularization and proposed the first federated learning compositional algorithm and analysis. [8] formulated the model personalization problem in federated learning as a model-agnostic meta-learning problem. In personalized federated learning, each client's task assignment is fixed and there is no task sampling on each client in the training procedure. The sampling of the inner layer and outer layer are independent. Therefore, personalized federated learning is formulated as the stochastic compositional optimization [19]. [33] solves personalized federated learning utilizing SCGD, in contrast to SGD in [19], to reduce the convergence complexities. However, the algorithm in [33] has a drawback in that keeping an inner state for each task is necessary, which is prohibitively expensive in large-scale settings. More recently, [11] proposed a momentum-like method for nonconvex problems with better complexities to solve the stochastic compositional problem in the federated learning setting. Although [11] claims their algorithm can be used in the MAML problem, it does not consider the two fundamental characteristics in MAML, i.e., task sampling and the dependency of inner data distribution on the sampled task.

Overall, problems (1) and (2) differ in two aspects: i) in Problem (2), the inner randomness $\eta$ and the outer randomness $\xi$ are independent, while in Problem (1), $\eta$ is conditionally dependent on the $\xi$; and ii) in Problem (1), the inner function depends on both $\xi$ and $\eta$. Therefore, Problem (2) can be regarded as a special case of (1). Thus, the conditional stochastic optimization (1) is more general.

Table 1: Complexity summary of proposed federated conditional stochastic optimization algorithms to reach an $\varepsilon$-stationary point. Sample complexity is defined as the number of calls to the First-order Oracle (IFO) by clients to reach an $\varepsilon$-stationary point. Communication complexity denotes the total number of back-and-forth communication rounds between each client and the central server required to reach an $\varepsilon$-stationary point.

| Algorithm | Sample | Communication |
|---|---|---|
| FCSG | $O\left(\epsilon^{-6}\right)$ | $O\left(\epsilon^{-3}\right)$ |
| FCSG-M | $O\left(\epsilon^{-6}\right)$ | $O\left(\epsilon^{-3}\right)$ |
| Lower Bound [16] | $O\left(\epsilon^{-5}\right)$ | - |
| Acc-FCSG-M | $O\left(\epsilon^{-5}\right)$ | $O\left(\epsilon^{-2}\right)$ |

## 3 Preliminary

For solving the problem (1), we first consider the local objective $F^n(x)$ and its gradient. We have

$$F^n(x) = \mathbb{E}_{\xi^n} f_{\xi^n}^n(\mathbb{E}_{\eta^n|\xi^n} g_{\eta^n}^n(x, \xi^n))$$

$$\nabla F^n(x) = \mathbb{E}_{\xi^n}\left[(\mathbb{E}_{\eta^n|\xi^n}\nabla g_{\eta^n}^n(x, \xi^n))]^\top \nabla f_{\xi^n}^n(\mathbb{E}_{\eta^n|\xi^n} g_{\eta^n}^n(x, \xi^n))\right]$$

Since there are two layers of stochastic functions, the standard stochastic gradient estimator is not an unbiased estimation for the full gradient. Instead of constructing an unbiased stochastic gradient estimator, [17] considered a biased estimator of $\nabla F(x)$ using one sample of $\xi$ and $m$ samples of $\eta$:

$$\nabla \hat{F}^n\left(x; \xi^n, \mathcal{B}_n\right) = (\frac{1}{m}\sum_{\eta_j^n \in \mathcal{B}_n}\nabla g_{\eta_j^n}^n(x, \xi^n))^\top \nabla f_{\xi^n}^n(\frac{1}{m}\sum_{\eta_j^n \in \mathcal{B}_n} g_{\eta_j^n}^n(x, \xi^n))$$

where $\mathcal{B}_n = \left\{\eta_j^n\right\}_{j=1}^m$. And $\nabla \hat{F}^n\left(x; \xi^n, \mathcal{B}_n\right)$ is the gradient of an empirical objective such that

$$\hat{F}^n\left(x; \xi^n, \mathcal{B}_n\right) := f_{\xi^n}^n(\frac{1}{m}\sum_{\eta_j^n \in \mathcal{B}_n} g_{\eta_j^n}^n(x, \xi^n)) \tag{3}$$

### 3.1 Assumptions

**Assumption 3.1.** (Smoothness) $\forall n \in [N]$, the function $f_{\xi^n}^n(\cdot)$ is $S_f$-Lipschitz smooth, and the function $g_{\eta^n}^n(\cdot, \xi^n)$ is $S_g$-Lipschitz smooth, *i.e.*, for a sample $\xi^n$ and $m$ samples $\{\eta_j^n\}_{j=1}^m$ from the conditional distribution $P(\eta^n \mid \xi^n)$, $\forall x_1, x_2 \in \text{dom } f^n(\cdot)$, and $\forall y_1, y_2 \in \text{dom } g^n(\cdot)$, there exist $S_f > 0$ and $S_g > 0$ such that

$$\mathbb{E}\|\nabla f_{\xi^n}^n(x_1) - \nabla f_{\xi^n}^n(x_2)\| \le S_f\|x_1 - x_2\| \quad \mathbb{E}\|\nabla g_{\eta^n}^n(y_1, \xi^n) - \nabla g_{\eta^n}^n(y_2, \xi^n)\| \le S_g\|y_1 - y_2\|$$

Assumption 3.1 is a widely used assumption in optimization analysis. Many single-machine stochastic algorithms use this assumption, such as BSGD [17], SPIDER [9], STORM [4], ADSGD [1], and D2SG [12]. In distributed learning, the convergence analysis of distributed learning algorithms, such as DSAL [2], and many federated learning algorithms such as MIME [21], Fed-GLOMO [6], STEM [23] and FAFED [39] also depend on it.

**Assumption 3.2.** (Bounded gradient) $\forall n \in [N]$, the function $f^n(\cdot)$ is $L_f$-Lipschitz continuous, and the function $g^n(\cdot)$ is $L_g$-Lipschitz continuous, *i.e.*, $\forall x \in \text{dom } f^n(\cdot)$, and $\forall y \in \text{dom } g^n(\cdot)$, the second moments of functions are bounded as below:

$$\mathbb{E}\|\nabla f_{\xi^n}^n(x)\|^2 \le L_f^2 \quad \mathbb{E}\|\nabla g_{\eta^n}^n(y_1, \xi^n)\|^2 \le L_g^2$$

Assumption 3.2 is a typical assumption in the multi-layer problem optimization to constrain the upper bound of the gradient of each layer, as in [36, 28, 11, 14].

**Assumption 3.3.** (Bounded variance) [17] $\forall n \in [N]$, and $x \in \mathcal{X}$:

$$\sup_{\xi^n, x \in \mathcal{X}} \mathbb{E}_{\eta^n|\xi^n}\|g_{\eta^n}^n(x, \xi^n) - \mathbb{E}_{\eta^n|\xi^n} g_{\eta^n}^n(x, \xi^n)\|^2 \le \sigma_g^2$$

where $\sigma_g^2 < +\infty$. Assumption 3.3 indicates that the random vector $g_{\eta^n}$ has bounded variance.

**Algorithm 1** FCSG and FCSG-M Algorithm

---

1: **Input:** Parameters: $T$, momentum weight $\beta$, learning rate $\alpha$, the number of local updates $q$, inner batch size $m$ and outer batch size $b$, as well as the initial outer batch size B ;

2: **Initialize:** $x_0^n = \bar{x}_0 = \frac{1}{N}\sum_{k=1}^N x_0^n$. Draw $B$ samples of $\{\xi_{t,1}^n, \cdots, \xi_{t,B}^n\}$ and draw $m$ samples $\mathcal{B}_{0,i}^n = \{\eta_{ij}^n\}_{j=1}^m$ from $P(\eta^n \mid \xi_{0,i}^n)$ for each $\xi_{0,i}^n \in \{\xi_{t,1}^n, \cdots, \xi_{t,B}^n\}$; $u_1^n = \frac{1}{B}\sum_{i=1}^B \nabla\hat{F}^n(x_0^n; \xi_{0,i}^n, \mathcal{B}_{0,i}^n)$.

3: **for** $t = 1, 2, \ldots, T$ **do**

4:    **for** $n = 1, 2, \ldots, N$ **do**

5:       **if** $\mod (t, q) = 0$ **then**

6:          **Server Update:**

7:          $u_t^n = \bar{u}_t = \frac{1}{N}\sum_{i=1}^N u_t^n$

8:          $x_t^n = \bar{x}_t = \frac{1}{N}\sum_{n=1}^N (x_{t-1}^n - \alpha u_t^n)$

9:       **else**

10:         $x_t^n = x_{t-1}^n - \alpha u_t^n$

11:       **end if**

12:       Draw $b$ samples of $\{\xi_{t,1}^n, \cdots, \xi_{t,b}^n\}$

13:       Draw $m$ samples $\mathcal{B}_{t,n}^n = \{\eta_{ij}^n\}_{j=1}^m$ from $P(\eta^n \mid \xi_{t,i}^n)$ for each $\xi_{t,i}^n \in \{\xi_{t,1}^n, \cdots, \xi_{t,b}^n\}$,

14:       $u_{t+1}^n = \frac{1}{b}\sum_{i=1}^b \nabla\hat{F}^n(x_t^n; \xi_{t,i}^n, \mathcal{B}_{t,i}^n)$

15:       $u_{t+1}^n = (1-\beta)u_t^n + \frac{\beta}{b}\sum_{i=1}^b \nabla\hat{F}^n(x_t^n; \xi_{t,i}^n, \mathcal{B}_{t,i}^n)$

16:    **end for**

17: **end for**

18: **Output:** $x$ chosen uniformly random from $\{\bar{x}_t\}_{t=1}^T$.

---

## 4 Proposed Algorithms

In the section, we propose a class of federated first-order methods to solve the Problem (1). We first design a federated conditional stochastic gradient (FCSG) algorithm with a biased gradient estimator and the momentum-based algorithm FCSG-M. To further accelerate our algorithm and achieve the lower bound of sample complexity of the single-machine algorithm, we design the Acc-FCSG-M with a variance reduction technique. Table 1 summarizes the complex details of each algorithm.

### 4.1 Federated Conditional Stochastic Gradient (FCSG)

First, we design a federated conditional stochastic gradient (FCSG) algorithm with the biased gradient estimator. We leverage a mini-batch of conditional samples to construct the gradient estimator $u_t$ with controllable bias as (6). At each iteration, clients update their local models $x_t$ with local data, which can be found in Line 9-14 of Algorithm 1. Once every $q$ local iterations, the server collects local models and returns the averaged models to each client, as Line 5-8 of Algorithm 1. Here, the number of local update steps $q$ is greater than 1 such that the number of communication rounds is reduced to $T/q$. The details of the method are summarized in Algorithm 1. Then we study the convergence properties of our new algorithm FCSG. Detailed proofs are provided in the supplementary materials.

**Theorem 4.1.** *Suppose Assumptions 3.1, 3.2 and 3.3 hold, if $\alpha \leq \frac{1}{6qS_F}$, FCSG has the following convergence result*

$$\frac{1}{T}\sum_{t=0}^{T-1} \|\nabla F(\bar{x}_t)\|^2 \leq \frac{2[F(\bar{x}_0) - F(\bar{x}_T)]}{\alpha T} + \frac{2L_g^2 S_f^2 \sigma_g^2}{m} + \frac{2\alpha S_F L_f^2 L_g^2}{N} + 42(q-1)q\alpha^2 L_f^2 L_g^2 S_F^2$$

**Corollary 4.2.** *We choose $\alpha = \frac{1}{6S_F}\sqrt{\frac{N}{T}}$ and $q = (T/N^3)^{1/4}$, we have*

$$\frac{1}{T}\sum_{t=0}^{T-1} \|\nabla F(\bar{x}_t)\|^2 \leq \frac{12S_F[F(\bar{x}_0) - F(\bar{x}_*)]}{(NT)^{1/2}} + \frac{2L_g^2 S_f^2 \sigma_g^2}{m} + \frac{L_f^2 L_g^2}{6(NT)^{1/2}} + \frac{19L_f^2 L_g^2}{9(NT)^{1/2}}$$

*Remark* 4.3. We choose $B = b = O(1) \geq 1$, and $m = O(\varepsilon^{-2})$, according to Corollary 4.2 to let $\frac{1}{T}\sum_{t=0}^{T-1}\|\nabla F(\bar{x}_t)\|^2 \leq \varepsilon^2$, we get $T = O(N^{-1}\varepsilon^{-4})$. $O(N^{-1}\varepsilon^{-4})$ indicates the linear speedup of our algorithm. Given $q = (T/N^3)^{1/4}$, the communication complexity is $\frac{T}{q} = (NT)^{3/4} = O(\varepsilon^{-3})$. Then the sample complexity is $mT = O(N^{-1}\varepsilon^{-6})$.

## 4.2 Federated Conditional Stochastic Gradient with Momentum (FCSG-M)

Next, we propose a momentum-based local updates algorithm (FCSG-M) for federated conditional stochastic optimization problems. Momentum is a popular technique widely used in practice for training deep neural networks. The motivation behind it in local updates is to use the historic information (*i.e.*, averaging of stochastic gradients) to reduce the effect of stochastic gradient noise. The details of our method are shown in Algorithm 1.

Initially, each device utilizes the standard stochastic gradient descent method to update the model parameter, as seen in Line 2 of Algorithm 1. Afterward, compared with FCSG, at each step, each client uses momentum-based gradient estimators $u_t$ to update the local model, which can be found in Lines 9-15 of Algorithm 1. The coefficient $\beta$ for the update of $u_t$ should satisfy $0 < \beta < 1$. In every $q$ iterations, the clients communicate $\{x_t, u_t\}$ to the server, which computes the $\{\bar{x}_t, \bar{u}_t\}$, and returns the averaged model and gradient estimator to each client, as Lines 5-8 of Algorithm 1. Then we study the convergence properties of our new algorithm FCSG-M. The details of the proof are provided in the supplementary materials.

**Theorem 4.4.** *Suppose Assumptions 3.1, 3.2 and 3.3 hold, $\alpha \leq \frac{1}{6qS_F}$ and $\beta = 5S_F\eta$. FCSG-M has the following convergence result*

$$\frac{1}{T}\sum_{t=0}^{T-1}\mathbb{E}\|\nabla F(\bar{x}_t)\|^2 \leq 2\frac{F(\bar{x}_0) - F(\bar{x}_T)}{\alpha T}$$

$$+ \frac{96S_F^2}{5}q^2\alpha^2[L_f^2L_g^2(1 + \frac{1}{N}) + 3L_f^2L_g^2] + \frac{4L_g^2S_f^2\sigma_g^2}{m} + \frac{8L_f^2L_g^2}{\beta BT} + \frac{8\beta L_f^2L_g^2}{N}$$

**Corollary 4.5.** *We choose $\alpha = \frac{1}{6S_F}\sqrt{\frac{N}{T}}$, $q = (T/N^3)^{1/4}$, we have*

$$\frac{1}{T}\sum_{t=0}^{T-1}\|\nabla F(\bar{x}_t)\|^2 \leq \frac{12S_F[F(\bar{x}_0) - F(\bar{x}_*)]}{(NT)^{1/2}}$$

$$+ \frac{112L_f^2L_g^2}{3(NT)^{1/2}} + \frac{4L_g^2S_f^2\sigma_g^2}{m} + \frac{48L_f^2L_g^2}{5(NT)^{1/2}} + \frac{20L_f^2L_g^2}{3(NT)^{1/2}}$$

*Remark* 4.6. We choose $b = O(1)$, $B = O(1)$, and $m = O(\varepsilon^{-2})$. According to Corollary 4.5, to make $\frac{1}{T}\sum_{t=0}^{T-1}\|\nabla F(\bar{x}_t)\|^2 \leq \varepsilon^2$, we get $T = O(N^{-1}\varepsilon^{-4})$. Given $q = (T/N^3)^{1/4}$, the communication complexity is $\frac{T}{q} = (NT)^{3/4} = O(\varepsilon^{-3})$. The sample complexity is $mT = O(N^{-1}\varepsilon^{-6})$, which indicates FCSG-M also has the linear speedup with respect to the number of clients.

## 4.3 Acc-FCSG-M

In the single-machine setting, [17] presents that under the general nonconvex conditional stochastic optimization objective, the lower bound of sample complexity is $O(\varepsilon^{-5})$. It means that the sample complexity achieved by FCSG and FCSG-M could be improved for nonconvex smooth conditional stochastic optimization objectives. To match the above lower bound of sample complexity, we propose an accelerated version of the FCSG-M (Acc-FCSG-M) based on the momentum-based variance reduction technique. The details of the method are shown in Algorithm 2.

Similar to the FCSG-M, in the beginning, each client initializes the model parameters and utilizes the stochastic gradient descent method to calculate the gradient estimator. Subsequently, in every $q$ iterations, all clients perform communication with the central server, and the model parameters and gradient estimators are averaged. The key difference is that at Line 14 in Acc-FCSG-M, we use the momentum-based variance reduction gradient estimator $u_{t+1}^n$ to track the gradient and update the model. where $\beta \in (0,1)$. We establish the theoretical convergence guarantee of our new algorithm Acc-FCSG-M. All proofs are provided in the supplementary materials.

**Algorithm 2** Acc-FCSG-M Algorithm

---
1: **Input:** $T$, momentum weight $\beta$, learning rate $\alpha$, the number of local updates $q$, inner batch size $m$ and outer batch size $b$, as well as the initial outer batch size B ;
2: **Initialize:** $x_0^n = \frac{1}{N}\sum_{k=1}^N x_0^n$. Draw $B$ samples of $\{\xi_1^n, \cdots, \xi_B^n\}$ and draw $m$ samples $\mathcal{B}_{0,i}^n = \{\eta_{ij}^n\}_{j=1}^m$ from $P(\eta^n \mid \xi_i^n)$ for each $\xi_i^n \in \{\xi_1^n, \cdots, \xi_B^n\}$, then $u_1^n = \frac{1}{B}\sum_{i=1}^B \nabla \hat{F}^n(x_0^n; \xi_{0,i}^n, \mathcal{B}_{0,i}^n)$ for $n \in [N]$.
3: **for** $t = 1, 2, \ldots, T$ **do**
4:   **for** $n = 1, 2, \ldots, N$ **do**
5:     **if**   $\mathrm{mod}\,(t, q) = 0$ **then**
6:       **Server Update:**
7:         $u_t^n = \bar{u}_t = \frac{1}{N}\sum_{i=1}^N u_t^n$
8:         $x_t^n = \bar{x}_t = \frac{1}{N}\sum_{n=1}^N (x_{t-1}^n - \alpha u_t^n)$
9:     **else**
10:         $x_{t,n} = x_{t-1}^n - \alpha u_t^n$
11:     **end if**
12:     Draw $b$ samples of $\{\xi_{t,1}^n, \cdots, \xi_{t,b}^n\}$
13:     Draw $m$ samples $\mathcal{B}_{t,n}^n = \{\eta_{ij}^n\}_{j=1}^m$ from $P(\eta^n \mid \xi_{t,i}^n)$ for each $\xi_{t,i}^n \in \{\xi_{t,1}^n, \cdots, \xi_{t,B}^n\}$,
14:       $u_{t+1}^n = \frac{1}{b}\sum_{i=1}^b \nabla \hat{F}^n(x_t^n; \xi_{t,i}^n, \mathcal{B}_{t,i}^n) + (1-\beta)(u_t^n - \frac{1}{b}\sum_{i=1}^b \nabla \hat{F}^n(x_{t-1}^n; \xi_{t,i}^n, \mathcal{B}_{t,i}^n))$
15:   **end for**
16: **end for**
17: **Output:** $x$ chosen uniformly random from $\{\bar{x}_t\}_{t=1}^T$.

---

**Theorem 4.7.** *Suppose Assumptions 3.1, 3.2 and 3.3 hold, $\alpha \leq \frac{1}{6qS_F}$ and $\beta = 5S_F\alpha$. Acc-FCSG-M has the following*

$$\frac{1}{T}\sum_{t=0}^{T-1}\|\nabla F(\bar{x}_t)\|^2 \leq \frac{2[F(\bar{x}_0) - F(\bar{x}_T)]}{T\alpha} + \frac{3L_f^2 L_g^2}{\beta BNT} + \frac{13L_f^2 L_g^2 c^2}{6S_F^2}\alpha^2 + \frac{3L_g^2 S_f^2 \sigma_g^2}{m} + \frac{6\beta L_f^2}{Nb}$$

**Corollary 4.8.** *We choose $q = \left(T/N^2\right)^{1/3}$. Therefore, $\alpha = \frac{1}{12qS_F} = \frac{N^{2/3}}{12S_F T^{1/3}}$, since $c = \frac{30S_F^2}{bN}$, we have $\beta = c\alpha^2 = \frac{5N^{1/3}}{24T^{2/3}b}$. And let $B = \frac{T^{1/3}}{N^{2/3}}$. Therefore, we have*

$$\frac{1}{T}\sum_{t=0}^{T-1}\|\nabla F(\bar{x}_t)\|^2 \leq \frac{24S_F[F(\bar{x}_0) - F(\bar{x}_*)]}{(NT)^{2/3}}$$

$$+ \frac{72L_f^2 L_g^2 b}{5(NT)^{2/3}} + \frac{325L_f^2 L_g^2}{24b^2(TN)^{2/3}} + \frac{3L_g^2 S_f^2 \sigma_g^2}{m} + \frac{5L_f^2}{4b^2(NT)^{2/3}}$$

*Remark* 4.9. We choose b as $O(1)(b \geq 1)$ and $m = O(\varepsilon^{-2})$. According to Corollary 4.8 to make $\frac{1}{T}\sum_{t=0}^{T-1}\|\nabla F(\bar{x}_t)\|^2 \leq \varepsilon^2$, we get $T = O(N^{-1}\varepsilon^{-3})$ and $\frac{T}{q} = (NT)^{2/3} = O(\varepsilon^{-2})$. The sample complexity is $O(N^{-1}\varepsilon^{-5})$. The communication complexity is $O(\varepsilon^{-2})$. $T = O(N^{-1}\varepsilon^{-3})$ indicates the linear speedup of our algorithm.

## 5 Experiments

The experiments are run on CPU machines with AMD EPYC 7513 32-Core Processors as well as NVIDIA RTX A6000. The code is available [*] and Federated Online AUPRC maximization task follow [38] [*].

### 5.1 Invariant Logistic Regression

Invariant learning has an important role in robust classifier training [27]. In this section, We compare the performance of our algorithms, FCSG and FCSG-M, on the distributed invariant logistic regression

---
[*]https://github.com/xidongwu/Federated-Minimax-and-Conditional-Stochastic-Optimization/tree/main
[*]https://github.com/xidongwu/D-AUPRC

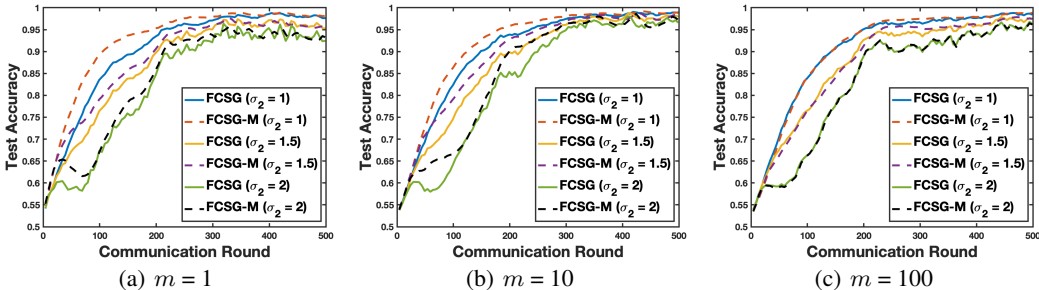

Figure 1: Test accuracy vs the number of communication rounds for different inner mini-batch ($m = 1, 10, 100$) under different noise ratios ($\sigma_2/\sigma_1 = 1, 1.5, 2$).

to evaluate the benefit from momentum and the effect of inner batch size, and the problem was formulated by [16]:

$$\min_x \frac{1}{N} \sum_{n=1}^{N} \mathbb{E}_{\xi^n = (a,b)} [l_n(x) + g(x)]$$

$$\text{where } l_n(x) = \log(1 + \exp(-b\mathbb{E}_{\eta^n|\xi^n}[(\eta^n)^\top x])) \quad g(x) = \lambda \sum_{i=1}^{d} \frac{\gamma x_i^2}{1 + \gamma x_i^2} \tag{4}$$

where $l_n(x)$ is the logistic loss function and $g(x)$ is a non-convex regularization. We follow the experimental protocols in [17] and set the dimension of the model as 10 over 16 clients. We construct the dataset $\xi^n = (a, b)$ and $\eta$ as follow: We sample $a \sim N(0, \sigma_1^2 I_d)$, set $b = \{\pm 1\}$ according to the sign of $a^\top x^*$, then sample $\eta_{ij}^n \sim N(a, \sigma_2^2 I_d)$. We choose $\sigma_1 = 1$, and consider the $\sigma_2/\sigma_1$ from $\{1, 1.5, 2\}$. At each local iteration, we use a fixed mini-batch size $m$ from $\{1, 10, 100\}$. The outer batch size is set as 1. We test the model with 50000 outer samples to report the test accuracy. We carefully tune hyperparameters for both methods. $\lambda = 0.001$ and $\alpha = 10$. We run a grid search for the learning rate and choose the learning rate in the set $\{0.01, 0.005, 0.001\}$. $\beta$ in FCSG-M are chosen from the set $\{0.001, 0.01, 0.1, 0.5, 0.9\}$. The local update step is set as 50. The experiments are presented in Figure 1.

Figure 1 shows that when the noise ratio $\sigma_2/\sigma_1$ increases, larger inner samples $m$ are needed, as suggested by the theory because a large batch size could reduce sample noise. In addition, when $m = 100$, FCSG and FCSG-M have similar performance. However, when batch size is small, compared with FCSG, FCSG-M has a more stable performance, because FCSG-M can use historic information to reduce the effect of stochastic gradient noise.

### 5.2 Federated Model-Agnostic Meta-Learning

Next, we evaluate our proposed algorithms with the few shot image classification task over the dataset with baselines: Local-SCGD and Local-SCGDM [11]. MOML [33] is not suitable for this task since MOML requires maintaining an inner state for each task which is not permitted due to the large number of tasks in the Omniglot dataset. Local-SCGD is the federated version of SCGD [36]. This task can be effectively solved via Model-Agnostic Meta-Learning [10].

Meta-learning aims to train a model on various learning tasks, such that the model can easily adapt to a new task using few training samples. Model-agnostic meta-learning (MAML) [10] is a popular meta-learning method to learn a good initialization with a gradient-based update, which can be formulated as the following conditional stochastic optimization problem:

$$\min_x \mathbb{E}_{i \sim \mathcal{P}_{\text{task}}, a \sim D_{\text{query}}^i} \mathcal{L}_i \left( \mathbb{E}_{b \sim D_{\text{support}}^i} (x - \lambda \nabla \mathcal{L}_i(x, b)), a \right)$$

where $\mathcal{P}_{\text{task}}$ denotes the learning tasks distribution, $D_{\text{support}}^i$ and $D_{\text{query}}^i$ are support (training) dataset and query (testing) dataset of the learning task $i$, respectively. $\mathcal{L}_i(\cdot, D_i)$ is the loss function on dataset $D_i$ of task $i$. And the $\lambda$ is a fixed meta step size. Assume $\xi = (i, a)$ and $\eta = b$, the MAML problem is an example of conditional stochastic optimization where the support (training) samples in the inner

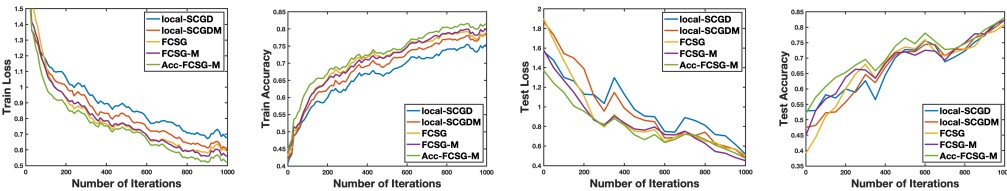

Figure 2: Convergence results of the 5-way-1-shot case over Omniglot Dataset.

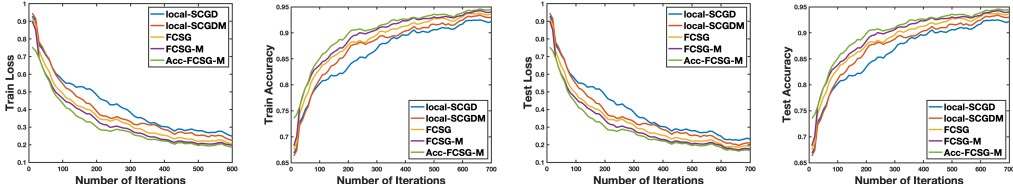

Figure 3: Convergence results of the 5-way-5-shot case over Omniglot Dataset.

layer for the meta-gradient update are drawn from the conditional distribution of $P(\eta \mid \xi)$ based on the sampled task in the outer layer.

Given there are a large number of pre-train tasks in MAML, federated learning is a good training strategy to improve efficiency because we can evenly distribute tasks over various clients and the global server coordinates clients to learn a good initial model collaboratively like MAML. Therefore, in Federated MAML, it is assumed that each device has part of the tasks. The optimization problem is defined as follows:

$$\min_{x \in \mathbb{R}^d} \frac{1}{N} \sum_{n=1}^{N} F^n(x) \triangleq \frac{1}{N} \sum_{n=1}^{N} f^n\left(g^n(x)\right) \tag{5}$$

$$\text{where } g^n(x) = \mathbb{E}_{\eta^n \sim \mathcal{D}_{i,\,\text{support}}^n}\left[x - \lambda \nabla \mathcal{L}_i^n\left(x; \eta^n\right)\right], f^n(y) = \mathbb{E}_{i \sim \mathcal{P}_{\text{task}}^n,\, a^n \sim \mathcal{D}_{i,\,\text{query}}^n} \mathcal{L}_i^n\left(y; a^n\right).$$

In this part, we apply our methods to few-shot image classification on the Omniglot [24, 10]. The Omniglot dataset contains 1623 different handwritten characters from 50 different alphabets and each of the 1623 characters consists of 20 instances drawn by different persons. We divide the characters to train/validation/test with 1028/172/423 by Torchmeta [7] and tasks are evenly partitioned into disjoint sets and we distribute tasks randomly among 16 clients. We conduct the fast learning of N-way-K-shot classification following the experimental protocol in [31]. The N-way-K-shot classification denotes we sample N unseen classes, randomly provide the model with K different instances of each class for training, and evaluate the model's ability to classify with new instances from the same N classes. We sample 15 data points for validation. We use a 4-layer convolutional neural model where each layer has 3 × 3 convolutions and 64 filters, followed by a ReLU nonlinearity and batch normalization [10]. The images from Omniglot are downsampled to 28 × 28. For all methods, the model is trained using a single gradient step with a learning rate of 0.4. The model was evaluated using 3 gradient steps [10]. Then we use grid search and carefully tune other hyper-parameters for each method. We choose the learning rate from the set $\{0.1, 0.05, 0.01\}$ and $\eta$ as 1 [11]. We select the inner state momentum coefficient for Local-SCGD and Local-SCGDM from $\{0.1, 0.5, 0.9\}$ and outside momentum coefficient for Local-SCGDM, FCSG-M and Acc-FCSG-M from $\{0.1, 0.5, 0.9\}$.

Figures 2 and 3 show experimental results in the 5-way-1-shot and 5-way-5-shot cases, respectively. Results show that our algorithms outperform baselines by a large margin. The main reason for Local-SCGD and Local-SCGDM to have bad performance is that converting the MAML optimization to the stochastic compositional optimization is unreasonable. It ignores the effect of task sampling on the training and inner training data distribution changes based on the sampled tasks in the outer layer. The use of the momentum-like inner state to deal with the MAML will slow down the convergence and we have to tune the extra momentum coefficient for the inner state. In addition, the momentum-like inner state also introduces extra communication costs because the server needs to average the inner state as in Local-SCGDM. In addition, comparing the results in Figures 2 and 3, we can see when

Table 2: Final averaged AP scores on the testing data.

| Datasets | FedAvg | CODA+ | FCSG | FCSG-M | Acc-FCSG-M |
|----------|--------|-------|------|--------|------------|
| MNIST    | 0.9357 | 0.9733 | 0.9868 | 0.9878 | 0.9879 |
| CIFAR-10 | 0.5059 | 0.6039 | 0.7130 | 0.7157 | 0.7184 |

the K increases in the few-shot learning, the training performance is improved, which matches the theoretical analysis that a large inner batch-size $m$ benefits the model training.

### 5.3 Federated Online AUPRC maximization

AUROC maximization in FL has been studied in [13, 42, 40] and AUPRC maximization is also used to solve the imbalanced classification. Existing AUPRC algorithms maintain an inner state for each data point. [38] consider the online AUPRC in the decentralized learning. For the large-scale distributed data over multiple clients, algorithms for online AUPRC maximization in FL is necessary.

Following [28] and [38], the surrogate function of average precision (AP) for online AUPRC maximization is:

$$\hat{AP} = \mathbb{E}_{\xi^+ \sim \mathcal{D}^+} \frac{\mathbb{E}_{\xi \sim \mathcal{D}} \mathbf{I}(y=1) \ell(x; z^+, z)}{\mathbb{E}_{\xi \sim \mathcal{D}} \ell(x; z^+, z)}$$

where $\ell(x; z^+, z) = (\max\{s - h(x; z^+) + h(x; z), 0\})^2$ and $h(x; z)$ is the prediction score function of input $z$ with model $x$. Federated Online AUPRC maximization could be reformulated as:

$$\min_x F(x) = \min_x \frac{1}{N} \sum_{n=1}^{n} \mathbb{E}_{\xi_n \sim \mathcal{D}_n^+} f(\mathbb{E}_{\xi_n' \sim \mathcal{D}_n} g^n(\mathbf{x}; \xi_n, \xi_n')) \qquad (6)$$

where $\xi_n = (z_n, y_n) \sim \mathcal{D}_n$ and $\xi_n^+ = (z_n^+, y_n^+) \sim \mathcal{D}_n^+$ are samples drawn from the whole datasets and positive datasets, respectively. It is a two-level problem and the inner objective depends on both $\xi$ and $\xi^+$. Since federated online AUPRC is a special example of a federated conditional stochastic optimization, our algorithms could be directly applied to it.

We choose MNIST dataset and CIFAR-10 datasets. As AUROC maximization in federated settings has been demonstrated in existing works [13, 42], we use CODA+ in [42] as a baseline. Another baseline is the FedAvg with cross-entropy loss. Since AUPRC is used for binary classification, the first half of the classes in the MNIST and CIFAR10 datasets are designated to be the negative class, and the rest half of the classes are considered to be the positive class. Then, we remove 80% of the positive examples in the training set to make it imbalanced, while keeping the test set unchanged. The results in Table 2 show that our algorithms could be used to solve the online AUPRC maximization in FL and it largely improves the model's performance.

## 6 Conclusion

In this paper, we studied federated conditional stochastic optimization under the general nonconvex setting. To the best of our knowledge, this is the first paper proposing algorithms for the federated conditional stochastic optimization problem. We first used the biased stochastic first-order gradient to design an algorithm called FCSG, which we proved to have a sample complexity of $O(\epsilon^{-6})$, and communication complexity of $O(\epsilon^{-3})$ to reach an $\epsilon$-stationary point. FCSG enjoys an appealing linear speedup with respect to the number of clients. To improve the empirical performances of FCSG, we also proposed a novel algorithm (*i.e.*, FCSG-M), which achieves the same theoretical guarantees as FCSG. To fill the gap from lower-bound complexity, we introduced an accelerated version of FCSG-M, called Acc-FCSG-M, using variance reduction technique, which is optimal for the nonconvex smooth federated conditional stochastic optimization problems as it matches the best possible complexity result achieved by the single-machine method. It has a sample complexity of $O(\epsilon^{-5})$, and communication complexity of $O(\epsilon^{-2})$. And the communication complexity achieves the best communication complexity of the nonconvex optimization in federated learning. Experimental results on the machine learning tasks validate the effectiveness of our algorithms.

**Limitation** Conditional stochastic optimization has much broader applications and a more comprehensive evaluation of our proposed algorithms on other use cases would be a promising future direction. In addition, FCSG-M and accelerated FCSG-M require higher communication overhead.

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
