# A Supplementary material

$s_t$ denotes the $s_t = \lfloor t/q \rfloor$. We define $g^n(x, \xi^n) = \mathbb{E}_{\eta^n | \xi^n} g^n_{\eta^n}(x, \xi^n)$ and $\hat{g}^n(x, \xi) = \frac{1}{m} \sum_{j=1}^m g^n(x, \xi; \eta_j), \hat{F}^n(x; \xi^n, \{\eta_j^n\}_{j=1}^m) = f^n_{\xi^n}(\hat{g}^n(x, \xi^n))$.

## A.1 Basic Lemma

For convenience, in the subsequent analysis, $s_t$ denotes the $s_t = \lfloor t/q \rfloor$ and $s_t \in [\lfloor T/q \rfloor]$. $g^n(x, \xi^n) = \mathbb{E}_{\eta^n | \xi^n} g^n_{\eta^n}(x, \xi^n)$ and $\hat{g}^n(x, \xi^n) = \frac{1}{m} \sum_{j=1}^m g^n(x, \xi^n; \eta_j), \hat{F}^n(x; \xi^n, \{\eta_j^n\}_{j=1}^m) = f^n_{\xi^n}(\hat{g}^n(x, \xi^n))$.

$$\nabla F^n(x) = \nabla \mathbb{E}_{\xi^n} \left[ f^n_{\xi^n}(g^n(x, \xi^n)) \right] = \mathbb{E}_{\xi^n} \left[ \nabla \left( f^n_{\xi^n}(g^n(x, \xi^n)) \right) \right]$$
$$= \mathbb{E}_{\xi^n} \left[ \nabla g^n(x, \xi^n)^\top \cdot \nabla_g f^n_{\xi^n}(g^n(x, \xi^n)) \right] \tag{7}$$

$$\hat{F}^n(x) = \mathbb{E}\hat{F}^n(x; \xi^n, \{\eta_j^n\}_{j=1}^m) = \mathbb{E}_{\xi^n} \mathbb{E}_{\eta^n | \xi^n} \left[ f^n_{\xi^n}(\hat{g}^n(x, \xi^n)) \right], \nabla \hat{F}^n(x)$$
$$= \mathbb{E}\nabla \hat{F}^n(x; \xi^n, \{\eta_j^n\}_{j=1}^m)$$

$$F(x) = \frac{1}{N} \sum_{n=1}^N F^n(x), \hat{F}(x) = \frac{1}{N} \sum_{n=1}^N \hat{F}^n(x) \tag{8}$$

**Lemma A.1.** *[23] For $x_t^n \in \mathbb{R}^d$, $n \in [N]$ and $\bar{x}_t \in \mathbb{R}^d$, we have*

$$\sum_{n=1}^N \|x_t^n - \bar{x}_t\|^2 \le \sum_{n=1}^N \|x_t^n\|^2 \tag{9}$$

**Lemma A.2.** *Under Assumption 3.1 3.2 and 3.3, in the n-th device, for a sample $\xi^n$ and $m$ i.i.d. samples $\{\eta_j^n\}_{j=1}^m$ from the conditional distribution $P(\eta^n \mid \xi^n)$, and $\forall x, x_1, x_2 \in \mathcal{X}$ that is independent of $\xi^n$ and $\{\eta_j^n\}_{j=1}^m$, we have*
*(a) (Lemma 2.2 in [16])*

$$\|\mathbb{E}\nabla \widehat{F}^n(x; \xi^n, \{\eta_j^n\}_{j=1}^m) - \nabla F^n(x)\|^2 \le \frac{L_g^2 S_f^2 \sigma_g^2}{m} \tag{10}$$

*(b) (Proposition B.1 in [17]) $F^n(x)$ and $\hat{F}^n(x)$ are $S_F$-Lipschitz smooth where $S_F = S_g L_f + S_f L_g^2$ and we also have*

$$\mathbb{E} \left\| \nabla \hat{F}^n \left( x_1; \xi, \{\eta_j\}_{j=1}^m \right) - \nabla \hat{F}^n \left( x_2; \xi, \{\eta_j\}_{j=1}^m \right) \right\| \le S_F \|x_1 - x_2\| \tag{11}$$

*(c) (Proposition B.1 in [17])*

$$\mathbb{E}_{\xi^n} \left\| \nabla f^n_{\xi^n}(y) - \nabla \mathbb{E}f^n_{\xi^n}(y) \right\|_2^2 \le L_f^2 \quad \mathbb{E}_{\xi^n | \eta^n} \left\| \nabla g^n_\eta(x, \xi^n) - \nabla \mathbb{E}g^n_\eta(x, \xi^n) \right\|_2^2 \le L_g^2$$

$$\mathbb{E}_{\xi^n} \mathbb{E}_{\xi^n | \eta^n} \left\| \nabla \left( f^n_{\xi^n}(\hat{g}^n(x, \xi^n)) \right) - \nabla \hat{F}^n(x) \right\|^2 \le L_f^2 L_g^2 \quad \left\| \nabla \hat{F}^n(x) \right\|^2 \le L_f^2 L_g^2$$

*(d) (Bounded Heterogeneity)*

$$\left\| \nabla \hat{F}^n(x) - \nabla \hat{F}^k(x) \right\|^2 \le 4L_f^2 L_g^2 \tag{12}$$

*Proof.* (e) Based on the (d), we know $\nabla \hat{F}^n(x)$ has bounded gradient, therefore,

$$\left\| \nabla \hat{F}^n(x) - \nabla \hat{F}^k(x) \right\|^2 \le 2\|\nabla \hat{F}^n(x)\|^2 + 2\|\nabla \hat{F}^k(x)\|^2 \le 4L_f^2 L_g^2$$

□

**Lemma A.3.** *For $n \in [N]$, we have*

$$\mathbb{E}\|\frac{1}{b} \sum_{i=1}^b \nabla \hat{F}^n(x_t^n; \xi_{t,i}^n, \mathcal{B}_{t,i}^n) - \nabla \hat{F}^n(x_t^n)\|^2 \le \frac{L_f^2 L_g^2}{b}$$

$$\sum_{n=1}^N \mathbb{E}\|\nabla \hat{F}^n(x_t^n) - \frac{1}{N} \sum_{k=1}^N \nabla \hat{F}^k(x_t^k)\|^2 \le 6S_F^2 \sum_{n=1}^N \mathbb{E}\|x_t^n - \bar{x}_t\|^2 + 12NL_f^2 L_g^2$$

*Proof.* 1) we have

$$\mathbb{E}\|\frac{1}{b}\sum_{i=1}^{b}\nabla\hat{F}^n(x_t^n;\xi_{t,i}^n,\mathcal{B}_{t,i}^n)-\nabla\hat{F}^n(x_t^n)\|^2=\frac{1}{b^2}\mathbb{E}\|\sum_{i=1}^{b}\nabla\hat{F}^n(x_t^n;\xi_{t,i}^n,\mathcal{B}_{t,i}^n)-\nabla\hat{F}^n(x_t^n)\|^2$$

$$=\frac{1}{b^2}\sum_{i=1}^{b}\mathbb{E}\|\nabla\hat{F}^n(x_t^n;\xi_{t,i}^n,\mathcal{B}_{t,i}^n)-\nabla\hat{F}^n(x_t^n)\|^2\leq\frac{L_f^2 L_g^2}{b}$$

where the second equality is due to $\mathbb{E}[\nabla\hat{F}^n(x_t^n;\xi_{t,i}^n,\mathcal{B}_{t,i}^n)-\nabla\hat{F}^n(x_t^n)]=0$ and the last inequality follows Lemma A.2 (c).

2)

$$\sum_{n=1}^{N}\mathbb{E}\|\nabla\hat{F}^n(x_t^n)-\frac{1}{N}\sum_{k=1}^{N}\nabla\hat{F}^k(x_t^k)\|^2$$

$$\leq 3\sum_{n=1}^{N}\mathbb{E}\left[\|\nabla\hat{F}^n(x_t^n)-\nabla\hat{F}^n(\bar{x}_t)\|^2+\|\nabla\hat{F}(\bar{x}_t)-\frac{1}{N}\sum_{k=1}^{N}\nabla\hat{F}(x_t^k)\|^2+\|\nabla\hat{F}^n(\bar{x}_t)-\nabla\hat{F}(\bar{x}_t)\|^2\right]$$

$$\leq 6S_F^2\sum_{n=1}^{N}\mathbb{E}\|x_t^n-\bar{x}_t\|^2+3\sum_{n=1}^{N}\mathbb{E}\|\nabla\hat{F}^n(\bar{x}_t)-\nabla\hat{F}(\bar{x}_t)\|^2$$

$$\leq 6S_F^2\sum_{n=1}^{N}\mathbb{E}\|x_t^n-\bar{x}_t\|^2+12NL_f^2 L_g^2 \tag{13}$$

where the second inequality is due to Assumption A.2 (b) and the last inequality is due to Assumption A.2 (d). $\qquad\square$

## B  Proof of FCSG Algorithm

### B.1  Proofs of the Intermediate Lemmas

**Lemma B.1.** *Assume the sequence $\{\bar{x}_t\}_{t=0}^{T-1}$ is generated from FCSG in Algorithm 1, if $\alpha\leq\frac{1}{6S_F q}$, we have*

$$\sum_{t=0}^{T-1}\frac{1}{N}\sum_{n=1}^{N}\mathbb{E}\left\|x_t^n-\bar{x}_t\right\|^2\leq 39(q-1)q\alpha^2 L_f^2 L_g^2 T \tag{14}$$

*Proof.* (1) if $t=s_t q$, we have

$$\sum_{n=1}^{N}\left\|x_{s_t q}^n-\bar{x}_{s_t q}\right\|^2=0 \tag{15}$$

(2) if $t>s_t q$, we have

$$x_t^n=x_{s_t q}^n-\sum_{s=s_t q+1}^{t}\alpha u_s^n \quad \bar{x}_t=\bar{x}_{s_t q}-\sum_{s=s_t q+1}^{t}\alpha\bar{u}_s \tag{16}$$

It implies that

$$\frac{1}{N}\sum_{n=1}^{N}\mathbb{E}\left\|x_t^n-\bar{x}_t\right\|^2=\frac{1}{N}\sum_{n=1}^{N}\mathbb{E}\left\|x_{s_t q}^n-\bar{x}_{s_t q}-\left(\sum_{s=s_t q+1}^{t}\alpha u_s^n-\sum_{s=s_t q+1}^{t}\alpha\bar{u}_s\right)\right\|^2$$

$$=\frac{1}{N}\sum_{n=1}^{N}\mathbb{E}\|\sum_{s=s_t q+1}^{t}\alpha(u_s^n-\bar{u}_s)\|^2$$

$$\leq\frac{(q-1)\alpha^2}{N}\sum_{s=s_t q+1}^{t}\sum_{n=1}^{N}\mathbb{E}\|u_s^n-\bar{u}_s\|^2 \tag{17}$$

Furthermore, based on the definition of $u_t^n$, we have

$$\frac{1}{N} \sum_{n=1}^{N} \mathbb{E} \|x_t^n - \bar{x}_t\|^2$$

$$\leq \frac{(q-1)\alpha^2}{N} \sum_{s=s_t q}^{t-1} \sum_{n=1}^{N} \mathbb{E} \left\| \frac{1}{b} \sum_{i=1}^{b} \nabla \hat{F}^n(x_t^n; \xi_{t,i}^n, \mathcal{B}_{t,i}^n) - \frac{1}{N} \sum_{k=1}^{N} \frac{1}{b} \sum_{i=1}^{b} \nabla \hat{F}^k(x_t^k; \xi_{t,i}^k, \mathcal{B}_{t,i}^k) \right\|^2$$

$$\overset{(a)}{\leq} \frac{2(q-1)\alpha^2}{N} \sum_{s=s_t q}^{t-1} \sum_{n=1}^{N} \mathbb{E} \left[ \left\| \left[ \frac{1}{b} \sum_{i=1}^{b} \nabla \hat{F}^n(x_t^n; \xi_{t,i}^n, \mathcal{B}_{t,i}^n) - \nabla \hat{F}^n(x_t^n) \right] \right. \right.$$

$$\left. \left. - \frac{1}{N} \sum_{k=1}^{N} \left[ \frac{1}{b} \sum_{i=1}^{b} \nabla \hat{F}^k(x_t^k; \xi_{t,i}^k, \mathcal{B}_{t,i}^k) - \nabla \hat{F}^k(x_t^k) \right] \right\|^2 + \left\| \left[ \nabla \hat{F}^n(x_t^n) - \frac{1}{N} \sum_{k=1}^{N} \nabla \hat{F}^k(x_t^k) \right] \right\|^2 \right]$$

$$\overset{(b)}{\leq} \frac{2(q-1)\alpha^2}{N} \sum_{s=s_t q}^{t-1} \left[ \sum_{n=1}^{N} \mathbb{E} \left\| \frac{1}{b} \sum_{i=1}^{b} \nabla \hat{F}^n(x_t^n; \xi_{t,i}^n, \mathcal{B}_{t,i}^n) - \nabla \hat{F}^n(x_t) \right\|^2 + \sum_{n=1}^{N} \mathbb{E} \|\hat{F}^n(x_t^n) - \hat{F}(x_t)\|^2 \right]$$

$$\overset{(c)}{\leq} \sum_{s=s_t q}^{t-1} [26(q-1)\alpha^2 L_f^2 L_g^2 + \frac{12(q-1)\alpha^2 S_F^2}{N} \sum_{n=1}^{N} \mathbb{E}\|x_t^n - \bar{x}_t\|^2]$$

$$\leq 26(q-1)q\alpha^2 L_f^2 L_g^2 + \frac{12(q-1)q\alpha^2 S_F^2}{N} \sum_{n=1}^{N} \mathbb{E}\|x_t^n - \bar{x}_t\|^2] \tag{18}$$

where (a) holds by $\|a + b\|^2 \leq 2\|a\|^2 + 2\|b\|^2$; the (b) holds due to Lemma A.1; the (c) holds by Lemma A.3. Summing both sides from $s = s_t q$ to $\bar{s}$ where $\bar{s} = [s_t q, (s_t + 1)q)$, we get

$$\sum_{s=s_t q}^{\bar{s}} \frac{1}{N} \sum_{n=1}^{N} \mathbb{E} \|x_s^n - \bar{x}_s\|^2 \leq \sum_{s=s_t q}^{\bar{s}} [26(q-1)q\alpha^2 L_f^2 L_g^2 + \frac{12(q-1)q\alpha^2 S_F^2}{N} \sum_{n=1}^{N} \mathbb{E}\|x_s^n - \bar{x}_s\|^2]$$

Rearranging the terms, we get

$$\left(1 - 12 S_F^2 q^2 \alpha^2\right) \sum_{s=s_t q}^{\bar{s}} \frac{1}{N} \sum_{n=1}^{N} \mathbb{E} \|x_s^n - \bar{x}_s\|^2 \leq \sum_{s=s_t q}^{\bar{s}} 26(q-1)q\alpha^2 L_f^2 L_g^2 \tag{19}$$

Finally, using the fact that $\alpha q \leq \frac{1}{6 S_F}$ we have $1 - 12 S_F^2 q^2 \alpha^2 \geq 2/3$. Multiplying, both sides by $3/2$ we get

$$\sum_{s=s_t q}^{\bar{s}} \frac{1}{N} \sum_{n=1}^{N} \mathbb{E} \|x_t^n - \bar{x}_t\|^2 \leq \sum_{s=s_t q}^{\bar{s}} 39(q-1)q\alpha^2 L_f^2 L_g^2 \tag{20}$$

Finally, when we consider the sum over t from 0 to T - 1, we get

$$\sum_{t=0}^{T-1} \frac{1}{N} \sum_{n=1}^{N} \mathbb{E} \|x_t^n - \bar{x}_t\|^2 \leq 39(q-1)q\alpha^2 L_f^2 L_g^2 T \tag{21}$$

$\square$

**Lemma B.2.** *Suppose the sequence $\{x_t\}_{t=0}^{T}$ be generated from FCSG in Algorithms 1. We have*

$$\mathbb{E}F(\bar{x}_{t+1}) \leq \mathbb{E}F(\bar{x}_t) - \frac{\alpha}{2}\mathbb{E}\|\nabla F(\bar{x}_t)\|^2 + \frac{\alpha S_F^2}{N} \sum_{n=1}^{N} \|x_t^n - \bar{x}_t\|^2 + \frac{\alpha L_g^2 S_f^2 \sigma_g^2}{m} + \frac{\alpha^2 S_F L_f^2 L_g^2}{N}$$

*Proof.*

$$\mathbb{E}F(\bar{x}_{t+1})$$

$$\overset{(a)}{\leq} \mathbb{E}F(\bar{x}_t) + \mathbb{E}\langle \nabla F(\bar{x}_t), \bar{x}_{t+1} - \bar{x}_t \rangle + \frac{S_F}{2}\mathbb{E}\|\bar{x}_{t+1} - \bar{x}_t\|^2$$

$$\overset{(b)}{\leq} \mathbb{E}F(\bar{x}_t) - \alpha\mathbb{E}\langle \nabla F(\bar{x}_t), \bar{u}_{t+1} \rangle + \frac{\alpha^2 S_F}{2}\mathbb{E}\|\bar{u}_{t+1}\|^2$$

$$\overset{(c)}{=} \mathbb{E}F(\bar{x}_t) - (\frac{\alpha}{2} - \alpha^2 S_F)\|\frac{1}{N}\sum_{n=1}^{N}\nabla\hat{F}^n(x_t^n)\|^2 - \frac{\alpha}{2}\|\nabla F(\bar{x}_t)\|^2 + \frac{\alpha}{2}\mathbb{E}\|\nabla F(\bar{x}_t) - \frac{1}{N}\sum_{n=1}^{N}\nabla\hat{F}^n(x_t^n)\|^2$$

$$+ \alpha^2 S_F\mathbb{E}\|\bar{u}_{t+1} - \frac{1}{N}\sum_{n=1}^{N}\nabla\hat{F}^n(x_t^n)\|^2$$

$$\overset{(d)}{\leq} \mathbb{E}F(\bar{x}_t) - \frac{\alpha}{2}\mathbb{E}\|\nabla F(\bar{x}_t)\|^2 + \frac{\alpha}{2}\mathbb{E}\|\nabla F(\bar{x}_t) - \frac{1}{N}\sum_{n=1}^{N}\nabla\hat{F}^n(x_t^n)\|^2 + \alpha^2 S_F\mathbb{E}\|\bar{u}_{t+1} - \frac{1}{N}\sum_{n=1}^{N}\nabla\hat{F}^n(x_t^n)\|^2$$

$$\leq \mathbb{E}F(\bar{x}_t) - \frac{\alpha}{2}\mathbb{E}\|\nabla F(\bar{x}_t)\|^2 + \alpha\mathbb{E}\|\nabla F(\bar{x}_t) - \frac{1}{N}\sum_{n=1}^{N}\nabla F^n(x_t^n)\|^2$$

$$+ \alpha\mathbb{E}\|\frac{1}{N}\sum_{n=1}^{N}\nabla F^n(x_t^n) - \frac{1}{N}\sum_{n=1}^{N}\nabla\hat{F}^n(x_t^n)\|^2 + \alpha^2 S_F\mathbb{E}\|\frac{1}{N}\sum_{n=1}^{N}\nabla\hat{F}^n(x_t^n) - \bar{u}_{t+1}\|^2 \qquad (22)$$

where inequality (a) holds by the smoothness of $F(x)$; equality (b) follows from update step in Step 9 of Algorithm 1; (c) uses the fact that $\langle a, b \rangle = \frac{1}{2}[\|a\|^2 + \|a\|^2 - \|a-b\|^2]$, $\|a+b\|^2 \leq 2\|a\|^2 + 2\|b\|^2$, and $\mathbb{E}[\bar{u}_{t+1}] = \nabla\hat{F}(x_t)$; (d) results from that $\alpha S_F \leq \frac{1}{2}$. Taking expectation on both sides and considering the last third term

$$\mathbb{E}\|\nabla F(\bar{x}_t) - \frac{1}{N}\sum_{n=1}^{N}\nabla F^n(x_t^n)\|^2 \leq \frac{1}{N}\sum_{n=1}^{N}\mathbb{E}\|\nabla F^n(\bar{x}_t) - \nabla F^n(x_t^n)\|^2$$

$$\leq \frac{S_F^2}{N}\sum_{n=1}^{N}\|x_t^n - \bar{x}_t\|^2 \qquad (23)$$

Considering the last second term and using Lemma A.2 (a), we have

$$\mathbb{E}\|\frac{1}{N}\sum_{n=1}^{N}\nabla F^n(x_t^n) - \frac{1}{N}\sum_{n=1}^{N}\nabla\hat{F}^n(x_t^n)\|^2$$

$$\leq \frac{1}{N}\sum_{n=1}^{N}\mathbb{E}\|\nabla F^n(x_t^n) - \nabla\hat{F}^n(x_t^n)\|^2 \leq \frac{L_g^2 S_f^2 \sigma_g^2}{m} \qquad (24)$$

For the last term, given that $\mathbb{E}\bar{u}_{t+1} = \frac{1}{N}\sum_{n=1}^{N}\nabla\hat{F}^n(x_t^n)$, we have

$$\mathbb{E}\|\frac{1}{N}\sum_{n=1}^{N}\nabla\hat{F}^n(x_t^n) - \bar{u}_{t+1}\|^2 \leq \frac{L_f^2 L_g^2}{N} \qquad (25)$$

Therefore, we obtain the final result. □

## B.2 Proof of Theorem 4.1

Based on previous lemmas, we start to prove the convergence of Theorem 4.1.

*Proof.* Taking the telescoping sum of B.2 over $t$ from 0 to $T-1$,

$$\frac{1}{T}\sum_{t=0}^{T-1}\|\nabla F(\bar{x}_t)\|^2$$

$$\leq \frac{2[F(\bar{x}_0)-F(\bar{x}_T)]}{\alpha T} + \frac{2S_F^2}{TN}\sum_{t=0}^{T-1}\sum_{n=1}^{N}\|x_t^n-\bar{x}_t\|^2 + \frac{2L_g^2S_f^2\sigma_g^2}{m} + \frac{2\alpha S_F L_f^2 L_g^2}{N}$$

$$\leq \frac{2[F(\bar{x}_0)-F(\bar{x}_T)]}{\alpha T} + \frac{2L_g^2S_f^2\sigma_g^2}{m} + \frac{2\alpha S_F L_f^2 L_g^2}{N} + 78(q-1)q\alpha^2 L_f^2 L_g^2 S_F^2. \tag{26}$$

where the second inequality holds due to Lemma B.1. Furthermore, we choose $\alpha = \frac{1}{6S_F}\sqrt{\frac{N}{T}}$ and $q = (T/N^3)^{1/4}$, we have

$$\frac{1}{T}\sum_{t=0}^{T-1}\|\nabla F(\bar{x}_t)\|^2 \leq \frac{12 S_F[F(\bar{x}_0)-F(\bar{x}_*)]}{(NT)^{1/2}} + \frac{2L_g^2 S_f^2 \sigma_g^2}{m} + \frac{L_f^2 L_g^2}{6(NT)^{1/2}} + \frac{19 L_f^2 L_g^2}{9(NT)^{1/2}} \tag{27}$$

To let the right hand side less than $\varepsilon^2$ when $m = \varepsilon^{-2}$, we get $T = O(N^{-1}\varepsilon^{-4})$ and $\frac{T}{q} = (NT)^{3/4} = \varepsilon^{-3}$. $\qquad\square$

## C   Proof of FCSG-M Algorithm

### C.1   Proofs of the Intermediate Lemmas

**Lemma C.1.** *Suppose the sequence $\{x_t\}_{t=0}^T$ be generated from FCSG-M in Algorithms 1. We have*

$$\mathbb{E}F(\bar{x}_{t+1}) \leq \mathbb{E}F(\bar{x}_t) - (\frac{\alpha}{2}-\frac{\alpha^2 S_F}{2})\mathbb{E}\|\bar{u}_{t+1}\|^2 - \frac{\alpha}{2}\mathbb{E}\|\nabla F(\bar{x}_t)\|^2 + \frac{4\alpha S_F^2}{N}\sum_{n=1}^{N}\mathbb{E}\|x_t^n-\bar{x}_t\|^2$$

$$+ \frac{2\alpha L_g^2 S_f^2 \sigma_g^2}{m} + 2\alpha\mathbb{E}\|\nabla\hat{F}(\bar{x}_t)-\bar{u}_{t+1}\|^2 \tag{28}$$

*Proof.*

$$F(\bar{x}_{t+1})$$

$$\overset{(a)}{\leq} F(\bar{x}_t) + \langle\nabla F(\bar{x}_t),\bar{x}_{t+1}-\bar{x}_t\rangle + \frac{S_F}{2}\|\bar{x}_{t+1}-\bar{x}_t\|^2$$

$$\overset{(b)}{=} F(\bar{x}_t) - \alpha\langle\nabla F(\bar{x}_t),\bar{u}_{t+1}\rangle + \frac{\alpha^2 S_F}{2}\|\bar{u}_{t+1}\|^2$$

$$\overset{(c)}{=} F(\bar{x}_t) - (\frac{\alpha}{2}-\frac{\alpha^2 S_F}{2})\|\bar{u}_{t+1}\|^2 - \frac{\alpha}{2}\|\nabla F(\bar{x}_t)\|^2 + \frac{\alpha}{2}\|\nabla F(\bar{x}_t)-\bar{u}_{t+1}\|^2$$

$$\leq F(\bar{x}_t) - (\frac{\alpha}{2}-\frac{\alpha^2 S_F}{2})\|\bar{u}_{t+1}\|^2 - \frac{\alpha}{2}\|\nabla F(\bar{x}_t)\|^2 + \frac{4\alpha}{2}\|\nabla F(\bar{x}_t)-\frac{1}{N}\sum_{n=1}^{N}\nabla F^n(x_t^n)\|^2$$

$$+ \frac{4\alpha}{2}\|\frac{1}{N}\sum_{n=1}^{N}[\nabla F^n(x_t^n)-\nabla\hat{F}^n(x_t^n)]\|^2 + \frac{4\alpha}{2}\|\frac{1}{N}\sum_{n=1}^{N}\nabla\hat{F}^n(x_t^n)-\nabla\hat{F}(\bar{x}_t)\|^2$$

$$+ \frac{4\alpha}{2}\|\nabla\hat{F}(\bar{x}_t)-\bar{u}_{t+1}\|^2$$

where inequality (a) holds by the smoothness of $F(x)$; equality (b) follows from update step in Step 9 of FCSG-M in Algorithm 1; (c) uses the fact that $\langle a,b\rangle = \frac{1}{2}[\|a\|^2+\|a\|^2-\|a-b\|^2]$. Taking

expectation on both sides and give the fact that

$$\mathbb{E}\|\nabla F(\bar{x}_t) - \frac{1}{N}\sum_{n=1}^{N}\nabla F^n(x_t^n)\|^2 \leq \frac{1}{N}\sum_{n=1}^{N}\mathbb{E}\|\nabla F^n(\bar{x}_t) - \nabla F^n(x_t^n)\|^2 \leq \frac{S_F^2}{N}\sum_{n=1}^{N}\mathbb{E}\|x_t^n - \bar{x}_t\|^2$$

$$\mathbb{E}\|\frac{1}{N}\sum_{n=1}^{N}\nabla F^n(x_t) - \frac{1}{N}\sum_{n=1}^{N}\nabla\hat{F}^n(x_t)\|^2 \leq \frac{1}{N}\sum_{n=1}^{N}\mathbb{E}\|\nabla F^n(x_t) - \nabla\hat{F}^n(x_t)\|^2 \overset{(a)}{\leq} \frac{L_g^2 S_f^2 \sigma_g^2}{m}$$

$$\mathbb{E}\|\frac{1}{N}\sum_{n=1}^{N}\nabla\hat{F}^n(x_t^n) - \nabla\hat{F}(\bar{x}_t)\|^2 \leq \frac{S_F^2}{N}\sum_{n=1}^{N}\mathbb{E}\|x_t^n - \bar{x}_t\|^2$$

where (a) holds due to Lemma A.2 (a). Therefore, taking the expectation on both sides, we obtain the final results. $\qquad\square$

**Lemma C.2.** *Suppose that the sequence $\{u_t\}_{t=1}^T$ be generated from FCSG-M in Algorithm 1, we have*

$$\frac{1}{T}\sum_{t=0}^{T-1}\mathbb{E}\left\|\bar{u}_{t+1} - \nabla\hat{F}(\bar{x}_t)\right\|^2$$

$$\leq \frac{2}{\beta T}\mathbb{E}\left\|\bar{u}_1 - \nabla\hat{F}(\bar{x}_0)\right\|^2 + \frac{4\alpha^2 S_F^2}{\beta^2 T}\sum_{t=0}^{T-1}\mathbb{E}\|\bar{u}_{t+1}\|^2 + \frac{2\beta L_f^2 L_g^2}{N} + \frac{2S_F^2}{NT}\sum_{t=0}^{T-1}\sum_{n=1}^{N}\mathbb{E}\|x_t^n - \bar{x}_t\|^2$$

*Proof.* Recall that $\bar{u}_{t+1} = \frac{1}{N}\sum_{n=1}^{N}[\frac{\beta}{b}\sum_{i=1}^{b}\nabla\hat{F}^n(x_t^n;\xi_{t,i}^n,\mathcal{B}_{t,i}^n) + (1-\beta)u_t^n]$,

$$\mathbb{E}\left\|\bar{u}_{t+1} - \nabla\hat{F}(\bar{x}_t)\right\|^2 = \mathbb{E}\left\|\frac{\beta}{N}\sum_{n=1}^{N}\frac{1}{b}\sum_{i=1}^{b}\nabla\hat{F}^n(x_t^n;\xi_{t,i}^n,\mathcal{B}_{t,i}^n) + (1-\beta)\bar{u}_t - \nabla\hat{F}(\bar{x}_t)\right\|^2$$

$$= \mathbb{E}\|\nabla\hat{F}(\bar{x}_t) - (1-\beta)\bar{u}_t - \beta\frac{1}{N}\sum_{n=1}^{N}\nabla\hat{F}^n(\mathbf{x}_t^n) - \beta\frac{1}{N}\sum_{n=1}^{N}[\frac{1}{b}\sum_{i=1}^{b}\nabla\hat{F}^n(x_t^n;\xi_{t,i}^n,\mathcal{B}_{t,i}^n) - \nabla\hat{F}^n(\mathbf{x}_t^n)]\|^2$$

$$\overset{(a)}{=} \mathbb{E}\left\|(1-\beta)(\nabla\hat{F}(\bar{x}_t) - \bar{u}_t) + \beta\left(\nabla\hat{F}(\bar{x}_t) - \frac{1}{N}\sum_{n=1}^{N}\nabla\hat{F}^n(x_t^n)\right)\right\|^2$$

$$+ \beta^2\mathbb{E}\left\|\frac{1}{N}\sum_{n=1}^{N}[\frac{1}{b}\sum_{i=1}^{b}\nabla\hat{F}^n(x_t^n;\xi_{t,i}^n,\mathcal{B}_{t,i}^n) - \nabla\hat{F}^n(\mathbf{x}_t^n)]\right\|^2$$

$$\overset{(b)}{\leq} (1+c_1)(1-\beta)^2\mathbb{E}\left\|\nabla\hat{F}(\bar{x}_t) - \bar{u}_t\right\|^2 + \beta^2\left(1+\frac{1}{c_1}\right)\mathbb{E}\left\|\nabla\hat{F}(\bar{x}_t) - \frac{1}{N}\sum_{n=1}^{N}\nabla\hat{F}^n(x_t^n)\right\|^2$$

$$+ \beta^2\frac{L_f^2 L_g^2}{N}$$

Here, (a) holds due to the definition of $\nabla\hat{F}^n(x_t^n)$. (b) holds due to Young's inequality and A.2 (c). We choose $c_1 = \frac{\beta}{1-\beta}$, and $(1-\beta)(1+c_1) = 1$ and $(1+\frac{1}{c_1})\beta = 1$. Therefore, we have

$$\mathbb{E}\left\|\bar{u}_{t+1} - \nabla\hat{F}\left(\bar{x}_t\right)\right\|^2$$

$$\leq (1-\beta)\,\mathbb{E}\left\|\nabla\hat{F}(\bar{x}_t) - \nabla\hat{F}(\bar{x}_{t-1}) + \nabla\hat{F}(\bar{x}_{t-1}) - \bar{u}_t\right\|^2 + \frac{\beta}{N}\sum_{n=1}^N S_F^2\,\mathbb{E}\left\|x_t^n - \bar{x}_t\right\|^2 + \frac{\beta^2 L_f^2 L_g^2}{N}$$

$$\leq (1-\beta)\left[(1+c_2)\,\mathbb{E}\left\|\bar{u}_t - \nabla\hat{F}(\bar{x}_{t-1})\right\|^2 + \left(1+\frac{1}{c_2}\right)\mathbb{E}\left\|\nabla\hat{F}(\bar{x}_t) - \nabla\hat{F}(\bar{x}_{t-1})\right\|^2\right] + \frac{\beta^2 L_f^2 L_g^2}{N}$$

$$+ \frac{\beta S_F^2}{N}\sum_{n=1}^N \mathbb{E}\left\|x_t^n - \bar{x}_t\right\|^2$$

$$\overset{(a)}{\leq} \left(1 - \frac{\beta}{2}\right)\mathbb{E}\left\|\bar{u}_t - \nabla\hat{F}(\bar{x}_{t-1})\right\|^2 + \frac{2}{\beta}S_F^2\,\mathbb{E}\left\|\bar{x}_t - \bar{x}_{t-1}\right\|^2 + \frac{\beta^2 L_f^2 L_g^2}{N} + \frac{\beta S_F^2}{N}\sum_{n=1}^N \mathbb{E}\left\|x_t^n - \bar{x}_t\right\|^2$$

$$= \left(1 - \frac{\beta}{2}\right)\mathbb{E}\left\|\bar{u}_t - \nabla\hat{F}(\bar{x}_{t-1})\right\|^2 + \frac{2\alpha^2 S_F^2}{\beta}\mathbb{E}\left\|\bar{u}_t\right\|^2 + \frac{\beta^2 L_f^2 L_g^2}{N} + \frac{\beta S_F^2}{N}\sum_{n=1}^N \mathbb{E}\left\|x_t^n - \bar{x}_t\right\|^2$$

where in inequality (a) we choose $c_2 = \frac{\beta}{2}$. Then, $(1-\beta)\left(1+\frac{\beta}{2}\right) \leq 1-\frac{\beta}{2}$, and $(1-\beta)\left(1+\frac{2}{\beta}\right) \leq \frac{2}{\beta}$. Then we have

$$\mathbb{E}\left\|\bar{u}_{t+1} - \nabla\hat{F}\left(\bar{x}_t\right)\right\|^2 - \mathbb{E}\left\|\bar{u}_t - \nabla\hat{F}(\bar{x}_{t-1})\right\|^2$$

$$\leq -\frac{\beta}{2}\mathbb{E}\left\|\bar{u}_t - \nabla\hat{F}(\bar{x}_{t-1})\right\|^2 + \frac{2\alpha^2 S_F^2}{\beta}\mathbb{E}\left\|\bar{u}_t\right\|^2 + \frac{\beta^2 L_f^2 L_g^2}{N} + \frac{\beta S_F^2}{N}\sum_{n=1}^N \mathbb{E}\left\|x_t^n - \bar{x}_t\right\|^2$$

By rearranging the terms and summing the t from 0 to $T-1$, one can get the final result. $\qquad\square$

**Lemma C.3.** *Assume $\alpha \leq \frac{1}{6qS_F}$ and the sequence $\{\bar{x}_t\}_{t=0}^{T-1}$ be generated from FCSG-M in Algorithm 1, the consensus error $\mathbb{E}\left\|x_t^n - \bar{x}_t\right\|^2$ satisfies*

$$\sum_{t=0}^{T-1}\frac{1}{N}\sum_{n=1}^N \mathbb{E}\left\|x_t^n - \bar{x}_t\right\|^2 \leq \frac{6q(q-1)\alpha^2 T}{5}[L_f^2 L_g^2(1+\frac{1}{N}) + 12L_f^2 L_g^2]$$

*Proof.* Based on the definition of $u_t$, we have

$$\frac{1}{N} \sum_{n=1}^{N} \|u_{t+1}^n - \bar{u}_{t+1}\|^2$$

$$= \frac{1}{N} \sum_{n=1}^{N} \mathbb{E} \left\| (1-\beta)(u_t^n - \bar{u}_t) + \beta[\frac{1}{b}\sum_{i=1}^{b} \nabla \hat{F}^n(x_t^n; \xi_{t,i}^n, \mathcal{B}_{t,i}^n) - \frac{1}{N}\sum_{k=1}^{N}\frac{1}{b}\sum_{i=1}^{b} \nabla \hat{F}^k(x_t^k; \xi_{t,i}^k, \mathcal{B}_{t,i}^k)] \right\|^2$$

$$\leq \frac{(1+c_3)(1-\beta)^2}{N} \sum_{n=1}^{N} \mathbb{E} \|u_t^n - \bar{u}_t\|^2$$

$$+ \left(1 + \frac{1}{c_3}\right) \frac{\beta^2}{N} \sum_{n=1}^{N} \mathbb{E} \left\| \frac{1}{b}\sum_{i=1}^{b} \nabla \hat{F}^n(x_t^n; \xi_{t,i}^n, \mathcal{B}_{t,i}^n) - \frac{1}{N}\sum_{k=1}^{N}\frac{1}{b}\sum_{i=1}^{b} \nabla \hat{F}^k(x_t^k; \xi_{t,i}^k, \mathcal{B}_{t,i}^k) \right\|^2$$

$$\overset{(a)}{=} \frac{(1-\beta)}{N} \sum_{n=1}^{N} \mathbb{E} \|u_t^n - \bar{u}_t\|^2 + \frac{\beta}{N} \sum_{n=1}^{N} \mathbb{E} \| \frac{1}{b}\sum_{i=1}^{b} \nabla \hat{F}^n(x_t^n; \xi_{t,i}^n, \mathcal{B}_{t,i}^n) - \nabla \hat{F}^n(x_t^n) + \nabla \hat{F}^n(x_t^n)$$

$$- \nabla \hat{F}^n(\bar{x}_t) + \nabla \hat{F}^n(\bar{x}_t) - \frac{1}{N}\sum_{k=1}^{N} \left[ \frac{1}{b}\sum_{i=1}^{b} \nabla \hat{F}^k(x_t^k; \xi_{t,i}^k, \mathcal{B}_{t,i}^k) - \nabla \hat{F}^k(x_t^k) + \nabla \hat{F}^k(x_t^k) \right]$$

$$+ \frac{1}{N}\sum_{k=1}^{n} \left[ \nabla \hat{F}^k(\bar{x}_t) - \nabla \hat{F}^k(\bar{x}_t) \right] \|^2$$

$$\leq \frac{(1-\beta)}{N} \sum_{n=1}^{N} \mathbb{E} \|u_t^n - \bar{u}_t\|^2 + \frac{\beta}{N} \sum_{n=1}^{N} \mathbb{E} \left[ \left\| \frac{1}{b}\sum_{i=1}^{b} \nabla \hat{F}^n(x_t^n; \xi_{t,i}^n, \mathcal{B}_{t,i}^n) - \nabla \hat{F}^n(x_t^n) \right\|^2 \right.$$

$$+ \left\| \frac{1}{N}\sum_{k=1}^{N} \left[ \frac{1}{b}\sum_{i=1}^{b} \nabla \hat{F}^k(x_t^k; \xi_{t,i}^k, \mathcal{B}_{t,i}^k) - \nabla \hat{F}^k(x_t^k) \right] \right\|^2 + \left\| \nabla \hat{F}^n(x_t^n) - \nabla \hat{F}^n(\bar{x}_t) + \nabla \hat{F}^n(\bar{x}_t) \right.$$

$$- \frac{1}{N}\sum_{k=1}^{N} \left[ \nabla \hat{F}^k(x_t^k) - \nabla \hat{F}^k(\bar{x}_t) \right] - \nabla \hat{F}(\bar{x}_t) \|^2 \right]$$

$$\leq \frac{(1-\beta)}{N} \sum_{n=1}^{N} \mathbb{E} \|u_t^n - \bar{u}_t\|^2 + \frac{\beta}{N} \sum_{n=1}^{N} \left[ L_f^2 L_g^2 + \frac{L_f^2 L_g^2}{N} + 3\mathbb{E} \left\| \nabla \hat{F}^n(x_t^n) - \nabla \hat{F}^n(\bar{x}_t) \right\|^2 \right.$$

$$+ 3\mathbb{E} \left\| \nabla \hat{F}^n(\bar{x}_t) - \nabla \hat{F}(\bar{x}_t) \right\|^2 + 3\mathbb{E} \left\| \nabla \hat{F}^n(x_t^n) - \nabla \hat{F}^n(\bar{x}_t) \right\|^2 \right]$$

$$\overset{(b)}{\leq} \frac{(1-\beta)}{N} \sum_{n=1}^{n} \mathbb{E} \|u_t^n - \bar{u}_t\|^2 + \beta[L_f^2 L_g^2(1+\frac{1}{N}) + 12 L_f^2 L_g^2] + \frac{6\beta S_F^2}{N} \sum_{n=1}^{N} \mathbb{E} \|x_t^n - \bar{x}_t\|^2 \qquad (29)$$

where (a) holds due to $c_3 = \frac{\beta}{1-\beta}$. So $(1-\beta)(1+c_3) = 1$, and $(1+\frac{1}{c_3})\beta = 1$; (b) holds due to Lemma A.2 (d). When $\mod(t,q) \neq 0$, using $u_{s_t q}^n = \bar{u}_{s_t q}$, we have

$$\frac{1}{N} \sum_{n=1}^{N} \|u_t^n - \bar{u}_t\|^2 \leq \sum_{s=s_t q}^{t-1} (1-\beta)^{t-1-s} \left[ \beta[L_f^2 L_g^2(1+\frac{1}{N}) + 12 L_f^2 L_g^2] + \frac{6\beta S_F^2}{N} \sum_{n=1}^{N} \mathbb{E} \|x_s^n - \bar{x}_s\|^2 \right]$$

$$(30)$$

Summing (30) from $t = s_t q + 1$ to $(s_t + 1)q$, we have

$$\sum_{t=s_t q+1}^{(s_t+1)q} \frac{1}{N} \sum_{n=1}^{N} \|u_t^n - \bar{u}_t\|^2$$

$$\leq \sum_{t=s_t q+1}^{(s_t+1)q-1} \sum_{s=s_t q}^{t-1} (1-\beta)^{t-1-s} \left[ \beta[L_f^2 L_g^2(1+\frac{1}{N}) + 12L_f^2 L_g^2] + \frac{6\beta S_F^2}{N} \sum_{n=1}^{N} \mathbb{E}\|x_s^n - \bar{x}_s\|^2 \right]$$

$$\leq \sum_{t=s_t q+1}^{(s_t+1)q-1} (\sum_{s=0}^{q} (1-\beta)^s) \left[ \beta[L_f^2 L_g^2(1+\frac{1}{N}) + 12L_f^2 L_g^2] + \frac{6\beta S_F^2}{N} \sum_{n=1}^{N} \mathbb{E}\|x_t^n - \bar{x}_t\|^2 \right]$$

$$\leq \sum_{t=s_t q+1}^{(s_t+1)q-1} \left[ [L_f^2 L_g^2(1+\frac{1}{N}) + 12L_f^2 L_g^2] + \frac{6 S_F^2}{N} \sum_{n=1}^{N} \mathbb{E}\|x_t^n - \bar{x}_t\|^2 \right] \tag{31}$$

where the last inequality holds due to $\sum_{s=0}^{q}(1-\beta)^s \leq \frac{1}{\beta}$. Similar to the (17), we have

$$\frac{1}{N} \sum_{n=1}^{N} \mathbb{E}\|x_t^n - \bar{x}_t\|^2 \leq \frac{(q-1)\alpha^2}{N} \sum_{s=s_t q+1}^{t} \sum_{n=1}^{N} \mathbb{E}\|u_s^n - \bar{u}_s\|^2$$

By summing $t$ from $s_t q + 1$ to $(s_t + 1)q$, we have

$$\sum_{t=s_t q+1}^{(s_t+1)q} \frac{1}{N} \sum_{n=1}^{N} \mathbb{E}\|x_t^n - \bar{x}_t\|^2$$

$$\leq \frac{(q-1)\alpha^2}{N} \sum_{t=s_t q+1}^{(s_t+1)q} \sum_{s=s_t q+1}^{t} \sum_{n=1}^{N} \mathbb{E}\|u_s^n - \bar{u}_s\|^2$$

$$\leq \frac{q(q-1)\alpha^2}{N} \sum_{t=s_t q+1}^{(s_t+1)q} \sum_{n=1}^{N} \mathbb{E}\|u_t^n - \bar{u}_t\|^2$$

$$\leq q(q-1)\alpha^2 \sum_{t=s_t q+1}^{(s_t+1)q-1} \left[ [L_f^2 L_g^2(1+\frac{1}{N}) + 12L_f^2 L_g^2] + \frac{6 S_F^2}{N} \sum_{n=1}^{N} \mathbb{E}\|x_t^n - \bar{x}_t\|^2 \right]$$

where the last inequality holds due to (31). Rearrange the terms in the above inequality, we have

$$\frac{1-6q(q-1)\alpha^2 S_F^2}{N} \sum_{t=s_t q+1}^{(s_t+1)q} \sum_{n=1}^{N} \mathbb{E}\|x_t^n - \bar{x}_t\|^2 \leq \sum_{t=s_t q+1}^{(s_t+1)q-1} q(q-1)\alpha^2 [L_f^2 L_g^2(1+\frac{1}{N}) + 12L_f^2 L_g^2]$$

Assume $\alpha \leq \frac{1}{6qS_F}$, we have $\frac{1-6q(q-1)\alpha^2 S_F^2}{N} \geq \frac{5}{6N}$, then by summing the t from 0 to T - 1, we get the final result.

$\square$

## C.2   Proof of Theorem 4.4

Based on aforementioned lemmas, we are ready to prove the convergence of Theorem

*Proof.* Based on the C.1, we have

$$\frac{1}{T}\sum_{t=0}^{T-1}\mathbb{E}\|\nabla F(\bar{x}_t)\|^2$$

$$\leq 2\frac{F(\bar{x}_0)-F(\bar{x}_T)}{\alpha T}-(1-\alpha S_F)\frac{1}{T}\sum_{t=0}^{T-1}\mathbb{E}\|\bar{u}_{t+1}\|^2+\frac{8S_F^2}{N}\frac{1}{T}\sum_{t=0}^{T-1}\sum_{n=1}^{N}\mathbb{E}\|x_t^n-\bar{x}_t\|^2$$

$$+\frac{4L_g^2S_f^2\sigma_g^2}{m}+\frac{4}{T}\sum_{t=0}^{T-1}\mathbb{E}\|\nabla\hat{F}(\bar{x}_t)-\bar{u}_{t+1}\|^2$$

$$\leq 2\frac{F(\bar{x}_0)-F(\bar{x}_T)}{\alpha T}-(1-\alpha S_F-\frac{16\alpha^2 S_F^2}{\beta^2})\frac{1}{T}\sum_{t=0}^{T-1}\mathbb{E}\|\bar{u}_{t+1}\|^2+\frac{16S_F^2}{N}\frac{1}{T}\sum_{t=0}^{T-1}\sum_{n=1}^{N}\mathbb{E}\|x_t^n-\bar{x}_t\|^2$$

$$+\frac{4L_g^2S_f^2\sigma_g^2}{m}+\frac{8}{\beta T}\mathbb{E}\left\|\bar{u}_1-\nabla\hat{F}(\bar{x}_0)\right\|^2+\frac{8\beta L_f^2L_g^2}{N}$$

$$\leq 2\frac{F(\bar{x}_0)-F(\bar{x}_T)}{\alpha T}-(1-\alpha S_F-\frac{16\alpha^2 S_F^2}{\beta^2})\frac{1}{T}\sum_{t=0}^{T-1}\|\bar{u}_{t+1}\|^2$$

$$+\frac{96S_F^2}{5}q(q-1)\alpha^2[L_f^2L_g^2(1+\frac{1}{N})+12L_f^2L_g^2]+\frac{4L_g^2S_f^2\sigma_g^2}{m}+\frac{8}{\beta T}\mathbb{E}\left\|\bar{u}_1-\nabla\hat{F}(\bar{x}_0)\right\|^2+\frac{8\beta L_f^2L_g^2}{N}$$

$$\leq 2\frac{F(\bar{x}_0)-F(\bar{x}_T)}{\alpha T}-(1-\alpha S_F-\frac{16\alpha^2 S_F^2}{\beta^2})\frac{1}{T}\sum_{t=0}^{T-1}\|\bar{u}_{t+1}\|^2$$

$$+\frac{96S_F^2}{5}q(q-1)\alpha^2[L_f^2L_g^2(1+\frac{1}{N})+12L_f^2L_g^2]+\frac{4L_g^2S_f^2\sigma_g^2}{m}+\frac{8L_f^2L_g^2}{\beta BT}+\frac{8\beta L_f^2L_g^2}{N}$$

where $(1-\alpha S_F-\frac{16\alpha^2 S_F^2}{\beta^2})>0$ when we choose $\alpha\leq\frac{1}{6qS_F}$ and $\beta=5S_F\alpha$. Finally, we have

$$\frac{1}{T}\sum_{t=0}^{T-1}\mathbb{E}\|\nabla F(\bar{x}_t)\|^2$$

$$\leq 2\frac{F(\bar{x}_0)-F(\bar{x}_T)}{\alpha T}+\frac{96S_F^2}{5}q^2\alpha^2[L_f^2L_g^2(1+\frac{1}{N})+12L_f^2L_g^2]+\frac{4L_g^2S_f^2\sigma_g^2}{m}+\frac{8L_f^2L_g^2}{\beta BT}+\frac{8\beta L_f^2L_g^2}{N}$$

We choose $b=O(1)$, $B=O(1)$, $\alpha=\frac{1}{6S_F}\sqrt{\frac{N}{T}}$, $q=(T/N^3)^{1/4}$, and $m=\varepsilon^{-2}$ , we have

$$\frac{1}{T}\sum_{t=0}^{T-1}\|\nabla F(\bar{x}_t)\|^2\leq\frac{12S_F[F(\bar{x}_0)-F(\bar{x}_*)]}{(NT)^{1/2}}+\frac{8[14L_f^2L_g^2]}{3(NT)^{1/2}}+\frac{4L_g^2S_f^2\sigma_g^2}{m}+\frac{48L_f^2L_g^2}{5(NT)^{1/2}}+\frac{20L_f^2L_g^2}{3(NT)^{1/2}}$$

To let the right hand is less than $\varepsilon^2$, we get $T=O(N^{-1}\varepsilon^{-4})$ and $\frac{T}{q}=(NT)^{3/4}=\varepsilon^{-3}$.

□

## D   Proof of Acc-FCSG-M Algorithm

### D.1   Proofs of the Intermediate Lemmas

**Lemma D.1.** *Suppose the sequence* $\{x_t\}_0^T$ *be generated from Acc-FCSG-M. We have*

$$F(\bar{x}_{t+1})\leq F(\bar{x}_t)-(\frac{\alpha}{2}-\frac{\alpha^2 S_F}{2})\|\bar{u}_{t+1}\|^2-\frac{\alpha}{2}\|\nabla F(\bar{x}_t)\|^2+\frac{3\alpha S_F^2}{2N}\|x_t^n-\bar{x}_t\|^2$$

$$+\frac{3\alpha L_g^2S_f^2\sigma_g^2}{2m}+\frac{3\alpha}{2}\|\frac{1}{N}\sum_{n=1}^{N}\nabla\hat{F}^n(x_t)-\bar{u}_{t+1}\|^2 \tag{32}$$

*Proof.*

$$F(\bar{x}_{t+1}) \overset{(a)}{\leq} F(\bar{x}_t) + \langle \nabla F(\bar{x}_t), \bar{x}_{t+1} - \bar{x}_t \rangle + \frac{S_F}{2} \|\bar{x}_{t+1} - \bar{x}_t\|^2$$

$$\overset{(b)}{=} F(\bar{x}_t) - \alpha\langle \nabla F(\bar{x}_t), \bar{u}_{t+1} \rangle + \frac{\alpha^2 S_F}{2} \|\bar{u}_{t+1}\|^2$$

$$\overset{(c)}{=} F(\bar{x}_t) - (\frac{\alpha}{2} - \frac{\alpha^2 S_F}{2})\|\bar{u}_{t+1}\|^2 - \frac{\alpha}{2}\|\nabla F(\bar{x}_t)\|^2 + \frac{\alpha}{2}\|\nabla F(\bar{x}_t) - \bar{u}_{t+1}\|^2$$

$$\leq F(\bar{x}_t) - (\frac{\alpha}{2} - \frac{\alpha^2 S_F}{2})\|\bar{u}_{t+1}\|^2 - \frac{\alpha}{2}\|\nabla F(\bar{x}_t)\|^2 + \frac{3\alpha}{2}\|\nabla F(\bar{x}_t) - \frac{1}{N}\sum_{n=1}^{N}\nabla F^n(x_t)\|^2$$

$$+ \frac{3\alpha}{2}\|\frac{1}{N}\sum_{n=1}^{N}\nabla F^n(x_t) - \frac{1}{N}\sum_{n=1}^{N}\nabla \hat{F}^n(x_t)\|^2 + \frac{3\alpha}{2}\|\frac{1}{N}\sum_{n=1}^{N}\nabla \hat{F}^n(x_t) - \bar{u}_{t+1}\|^2 \quad (33)$$

where inequality (a) holds by the smoothness of $F(x)$; equality (b) follows from update step in Step 9 of Acc-FCSG-M in Algorithm 2; (c) uses the fact that $\langle a, b \rangle = \frac{1}{2}[\|a\|^2 + \|a\|^2 - \|a - b\|^2]$. Taking expectation on both sides and considering the last third term

$$\mathbb{E}\|\nabla F(\bar{x}_t) - \frac{1}{N}\sum_{n=1}^{N}\nabla F^n(x_t)\|^2 \leq \frac{1}{N}\sum_{n=1}^{N}\mathbb{E}\|\nabla F^n(\bar{x}_t) - \nabla F^n(x_t)\|^2$$

$$\leq \frac{S_F^2}{N}\sum_{n=1}^{N}\|x_t^n - \bar{x}_t\|^2 \quad (34)$$

Considering the last second term, we have

$$\mathbb{E}\|\frac{1}{N}\sum_{n=1}^{N}\nabla F^n(x_t) - \frac{1}{N}\sum_{n=1}^{N}\nabla \hat{F}^n(x_t)\|^2 \leq \frac{1}{N}\sum_{n=1}^{N}\mathbb{E}\|\nabla F^n(x_t) - \nabla \hat{F}^n(x_t)\|^2$$

$$\leq \frac{L_g^2 S_f^2 \sigma_g^2}{m} \quad (35)$$

Therefore, we obtain

$$F(\bar{x}_{t+1}) \leq F(\bar{x}_t) - (\frac{\alpha}{2} - \frac{\alpha^2 S_F}{2})\|\bar{u}_{t+1}\|^2 - \frac{\alpha}{2}\|\nabla F(\bar{x}_t)\|^2 + \frac{3\alpha S_F^2}{2N}\|x_t^n - \bar{x}_t\|^2$$

$$+ \frac{3\alpha L_g^2 S_f^2 \sigma_g^2}{2m} + \frac{3\alpha}{2}\|\frac{1}{N}\sum_{n=1}^{N}\nabla \hat{F}^n(x_t) - \bar{u}_{t+1}\|^2 \quad (36)$$

$\square$

**Lemma D.2.** *(Lemma C.6 in [23]) Using the fact that $\mathbb{E}\bar{u}_{t+1} = \nabla\hat{F}(x_t)$ and $\bar{u}_{t+1}$ is the momentum-based variance reduction estimator, we have*

$$\mathbb{E}\|\bar{u}_{t+1} - \nabla\hat{F}(x_t)\|^2 \leq (1 - \beta)^2\mathbb{E}\|\bar{u}_t - \nabla\hat{F}(x_{t-1})\|^2 + \frac{8(1 - \beta)^2 S_F^2}{N^2 b}\frac{q - 1}{q}\sum_{n=1}^{N}\alpha^2\mathbb{E}\|u_t^n - \bar{u}_t\|^2$$

$$+ \frac{4(1 - \beta)^2 S_F^2 \alpha^2}{Nb}\mathbb{E}\|\bar{u}_t\|^2 + \frac{2\beta^2 L_f^2}{Nb} \quad (37)$$

**Lemma D.3.** *Assume that the sequences $u_t$ are generated from Acc-FCSG-M, set $\gamma = \frac{1}{q}$ and $\alpha \leq \frac{1}{12S_F q}$, and given that $\beta = c\alpha^2$, $c = \frac{30S_F^2}{bN}$ we have*

$$\frac{15}{72N}\sum_{t=s_t}^{\bar{s}}\alpha\sum_{n=1}^{N}\mathbb{E}\|u_t^n - \bar{u}_t\|^2 \leq \frac{1}{8}\sum_{t=s_t}^{\bar{s}}\alpha\mathbb{E}\|\bar{u}_t\|^2 + \left[\frac{L_f^2 c^2}{8bS_F^2} + \frac{3L_f^2 L_g^2 c^2}{8S_F^2}\right]\sum_{t=s_t}^{\bar{s}}\alpha^3 \quad (38)$$

*Proof.*

$$\sum_{n=1}^{N} \mathbb{E}\|u_{t+1}^n - \bar{u}_{t+1}\|^2 \leq \sum_{n=1}^{N} \mathbb{E}\left\| \frac{1}{b}\sum_{n=1}^{b} \nabla \hat{F}^n(x_t^n; \xi_{t,i}^n, \mathcal{B}_{t,i}^n) + (1-\beta)(u_t^n - \frac{1}{b}\sum_{n=1}^{b} \nabla \hat{F}^n(x_{t-1}^n; \xi_{t,i}^n, \mathcal{B}_{t,i}^n)) \right.$$

$$\left. - \frac{1}{N}\sum_{k=1}^{N}\left[ \frac{1}{b}\sum_{i=1}^{b} \nabla \hat{F}^k(x_t^k; \xi_{t,i}^k, \mathcal{B}_{t,i}^k) + (1-\beta)(u_t^k - \frac{1}{b}\sum_{i=1}^{b} \nabla \hat{F}^k(x_{t-1}^k; \xi_{t,i}^k, \mathcal{B}_{t,i}^k)) \right] \right\|^2$$

$$= \sum_{n=1}^{N} \mathbb{E}\left\| (1-\beta)(u_t^n - \bar{u}_t) + [\frac{1}{b}\sum_{i=1}^{b} \nabla \hat{F}^n(x_t^n; \xi_{t,i}^n, \mathcal{B}_{t,i}^n) - \frac{1}{N}\sum_{k=1}^{N}\frac{1}{b}\sum_{i=1}^{b} \nabla \hat{F}^k(x_t^k; \xi_{t,i}^k, \mathcal{B}_{t,i}^k) \right.$$

$$\left. - (1-\beta)[\frac{1}{b}\sum_{i=1}^{b} \nabla \hat{F}^n(x_{t-1}^n; \xi_{t,i}^n, \mathcal{B}_{t,i}^n) - \frac{1}{N}\sum_{k=1}^{N}\frac{1}{b}\sum_{i=1}^{b} \nabla \hat{F}^k(x_{t-1}^k; \xi_{t,i}^k, \mathcal{B}_{t,i}^k)]] \right\|^2$$

$$\leq (1+\gamma)(1-\beta)^2 \sum_{n=1}^{N} \mathbb{E}\|u_t^n - \bar{u}_t\|^2$$

$$+ (1+\frac{1}{\gamma})\mathbb{E}\left\| \left[ \frac{1}{b}\sum_{i=1}^{b} \nabla \hat{F}^n(x_t^n; \xi_{t,i}^n, \mathcal{B}_{t,i}^n) - \frac{1}{N}\sum_{k=1}^{N}\frac{1}{b}\sum_{i=1}^{b} \nabla \hat{F}^k(x_t^k; \xi_{t,i}^k, \mathcal{B}_{t,i}^k) \right] \right.$$

$$\left. - (1-\beta)\left[ \frac{1}{b}\sum_{i=1}^{b} \nabla \hat{F}^n(x_{t-1}^n; \xi_{t,i}^n, \mathcal{B}_{t,i}^n) - \frac{1}{N}\sum_{k=1}^{N}\frac{1}{b}\sum_{i=1}^{b} \nabla \hat{F}^k(x_{t-1}^k; \xi_{t,i}^k, \mathcal{B}_{t,i}^k) \right] \right\|^2 \tag{39}$$

where the second inequality is due to Young's inequality. For the second term, we have

$$\sum_{n=1}^{N} \mathbb{E}\left\| \left[ \frac{1}{b}\sum_{i=1}^{b} \nabla \hat{F}^n(x_t^n; \xi_{t,i}^n, \mathcal{B}_{t,i}^n) - \frac{1}{N}\sum_{k=1}^{N}\frac{1}{b}\sum_{i=1}^{b} \nabla \hat{F}^k(x_t^k; \xi_{t,i}^k, \mathcal{B}_{t,i}^k) \right] \right.$$

$$\left. - (1-\beta)\left[ \frac{1}{b}\sum_{i=1}^{b} \nabla \hat{F}^n(x_{t-1}^n; \xi_{t,i}^n, \mathcal{B}_{t,i}^n) - \frac{1}{N}\sum_{k=1}^{N}\frac{1}{b}\sum_{i=1}^{b} \nabla \hat{F}^n(x_{t-1}^n; \xi_{t,i}^n, \mathcal{B}_{t,i}^n) \right] \right\|^2$$

$$= \sum_{n=1}^{N} \mathbb{E}\left\| \left[ \frac{1}{b}\sum_{i=1}^{b} \nabla \hat{F}^n(x_t^n; \xi_{t,i}^n, \mathcal{B}_{t,i}^n) - \frac{1}{N}\sum_{k=1}^{N}\frac{1}{b}\sum_{i=1}^{b} \nabla \hat{F}^k(x_t^k; \xi_{t,i}^k, \mathcal{B}_{t,i}^k) \right] - \left[ \frac{1}{b}\sum_{i=1}^{b} \nabla \hat{F}^n(x_{t-1}^n; \xi_{t,i}^n, \mathcal{B}_{t,i}^n) \right. \right.$$

$$\left. \left. - \frac{1}{N}\sum_{k=1}^{n}\frac{1}{b}\sum_{i=1}^{b} \nabla \hat{F}^k(x_{t-1}^k; \xi_{t,i}^k, \mathcal{B}_{t,i}^k) \right] + \beta\left[ \frac{1}{b}\sum_{i=1}^{b} \nabla \hat{F}^n(x_{t-1}^n; \xi_{t,i}^n, \mathcal{B}_{t,i}^n) - \frac{1}{N}\sum_{k=1}^{N}\frac{1}{b}\sum_{i=1}^{b} \nabla \hat{F}^k(x_{t-1}^k; \xi_{t,i}^k, \mathcal{B}_{t,i}^k) \right] \right\|^2$$

$$\leq 2\sum_{n=1}^{N} \mathbb{E}\left\| \left[ \frac{1}{b}\sum_{i=1}^{b} \nabla \hat{F}^n(x_t^n; \xi_{t,i}^n, \mathcal{B}_{t,i}^n) - \frac{1}{N}\sum_{k=1}^{N}\frac{1}{b}\sum_{i=1}^{b} \nabla \hat{F}^k(x_t^k; \xi_{t,i}^k, \mathcal{B}_{t,i}^k) \right] \right.$$

$$\left. - \left[ \frac{1}{b}\sum_{i=1}^{b} \nabla \hat{F}^n(x_{t-1}^n; \xi_{t,i}^n, \mathcal{B}_{t,i}^n) - \frac{1}{N}\sum_{k=1}^{n}\frac{1}{b}\sum_{i=1}^{b} \nabla \hat{F}^k(x_{t-1}^k; \xi_{t,i}^k, \mathcal{B}_{t,i}^k) \right] \right\|^2$$

$$+ 2\beta^2 \sum_{n=1}^{N} \mathbb{E}\left\| \frac{1}{b}\sum_{i=1}^{b} \nabla \hat{F}^n(x_{t-1}^n; \xi_{t,i}^n, \mathcal{B}_{t,i}^n) - \frac{1}{N}\sum_{k=1}^{N}\frac{1}{b}\sum_{i=1}^{b} \nabla \hat{F}^k(x_{t-1}^k; \xi_{t,i}^k, \mathcal{B}_{t,i}^k) \right\|^2$$

$$\overset{(a)}{\leq} 2\sum_{n=1}^{N} \mathbb{E}\left\| \frac{1}{b}\sum_{i=1}^{b} \nabla \hat{F}^n(x_t^n; \xi_{t,i}^n, \mathcal{B}_{t,i}^n) - \frac{1}{b}\sum_{i=1}^{b} \nabla \hat{F}^n(x_{t-1}^n; \xi_{t,i}^n, \mathcal{B}_{t,i}^n) \right\|^2$$

$$+ 2\beta^2 \sum_{n=1}^{N} \mathbb{E}\| \frac{1}{b}\sum_{i=1}^{b} \nabla \hat{F}^n(x_{t-1}^n; \xi_{t,i}^n, \mathcal{B}_{t,i}^n) - \frac{1}{N}\sum_{k=1}^{N}\frac{1}{b}\sum_{i=1}^{b} \nabla \hat{F}^k(x_{t-1}^k; \xi_{t,i}^k, \mathcal{B}_{t,i}^k)\|^2$$

$$\overset{(b)}{\leq} 2S_F^2 \sum_{n=1}^{N} \mathbb{E}\|x_t^n - x_{t-1}^n\|^2 + 2\beta^2 \sum_{n=1}^{N} \mathbb{E}\left\| \frac{1}{b}\sum_{i=1}^{b} \nabla \hat{F}^n(x_{t-1}^n; \xi_{t,i}^n, \mathcal{B}_{t,i}^n) - \frac{1}{N}\sum_{k=1}^{N}\frac{1}{b}\sum_{i=1}^{b} \nabla \hat{F}^k(x_{t-1}^k; \xi_{t,i}^k, \mathcal{B}_{t,i}^k) \right\|^2 \tag{40}$$

where (a) holds due to Lemma A.1, and (b) uses Lemma A.2 (b). For the last term in (40), we have

$$
\sum_{n=1}^{N} \mathbb{E} \left\| \frac{1}{b} \sum_{i=1}^{b} \nabla \hat{F}^n(x_{t-1}^n; \xi_{t,i}^n, \mathcal{B}_{t,i}^n) - \frac{1}{N} \sum_{k=1}^{N} \frac{1}{b} \sum_{i=1}^{b} \nabla \hat{F}^k(x_{t-1}^k; \xi_{t,i}^k, \mathcal{B}_{t,i}^k) \right\|^2
$$

$$
= \sum_{n=1}^{N} \mathbb{E} \left\| \frac{1}{b} \sum_{i=1}^{b} \nabla \hat{F}^n(x_{t-1}^n; \xi_{t,i}^n, \mathcal{B}_{t,i}^n) - \nabla \hat{F}^n(x_{t-1}^n) - \frac{1}{N} \sum_{k=1}^{N} \left[ \frac{1}{b} \sum_{i=1}^{b} \nabla \hat{F}^k(x_{t-1}^k; \xi_{t,i}^k, \mathcal{B}_{t,i}^k) - \nabla F(x_{t-1}^k) \right] \right.
$$

$$
\left. + \left[ \nabla \hat{F}^n(x_{t-1}^n) - \frac{1}{N} \sum_{k=1}^{N} \nabla \hat{F}^k(x_{t-1}^k) \right] \right\|^2
$$

$$
\leq 2 \sum_{n=1}^{N} \mathbb{E} \| \frac{1}{b} \sum_{i=1}^{b} \nabla \hat{F}^n(x_{t-1}^n; \xi_{t,i}^n, \mathcal{B}_{t,i}^n) - \nabla \hat{F}^n(x_{t-1}^n) \|^2 + 2 \sum_{n=1}^{N} \mathbb{E} \| \hat{F}^n(x_{t-1}^n) - \frac{1}{N} \sum_{k=1}^{N} \nabla \hat{F}^k(x_{t-1}^k) \|^2
$$

$$
\leq 26 N L_f^2 L_g^2 + 12 S_F^2 \sum_{n=1}^{N} \mathbb{E} \| x_{t-1}^n - \bar{x}_{t-1} \|^2 \tag{41}
$$

where the first inequality is due to Lemma A.1 and the last inequality is due to Lemma A.2 (c) and Lemma A.3. Therefore, by combining above inequalities (39), (40) and (41), when $\mod(t+1, q) \neq 0$ we have

$$
\sum_{n=1}^{N} \mathbb{E} \| u_{t+1}^n - \bar{u}_{t+1} \|^2
$$

$$
\leq (1 - \beta)^2 (1 + \gamma) \sum_{n=1}^{N} \mathbb{E} \| u_t^n - \bar{u}_t \|^2 + 2 S_F^2 (1 + \frac{1}{\gamma}) \sum_{n=1}^{N} \mathbb{E} \| x_t^n - x_{t-1}^n \|^2
$$

$$
+ 52 N L_f^2 L_g^2 (1 + \frac{1}{\gamma}) \beta^2 + 24 S_F^2 (1 + \frac{1}{\gamma}) \beta^2 \sum_{n=1}^{N} \mathbb{E} \| x_{t-1}^n - \bar{x}_{t-1} \|^2
$$

$$
\overset{(a)}{\leq} (1 - \beta)^2 (1 + \gamma) \sum_{n=1}^{N} \mathbb{E} \| u_t^n - \bar{u}_t \|^2 + 2 S_F^2 (1 + \frac{1}{\gamma}) \sum_{n=1}^{N} \mathbb{E} \| \alpha u_t^n \|^2
$$

$$
+ 52 N L_f^2 L_g^2 (1 + \frac{1}{\gamma}) \beta^2 + 24 S_F^2 (1 + \frac{1}{\gamma}) \beta^2 (q - 1) \sum_{s=s_t q+1}^{t-1} \alpha^2 \sum_{n=1}^{N} \mathbb{E} \| u_s^n - \bar{u}_s \|^2
$$

$$
\leq (1 - \beta)^2 (1 + \gamma) \sum_{n=1}^{N} \mathbb{E} \| u_t^n - \bar{u}_t \|^2 + 4 S_F^2 (1 + \frac{1}{\gamma}) \sum_{n=1}^{N} \mathbb{E} \left[ \| \alpha(u_t^n - \bar{u}_t) \|^2 + \| \alpha \bar{u}_t \|^2 \right]
$$

$$
+ 52 N L_f^2 L_g^2 (1 + \frac{1}{\gamma}) \beta^2 + 24 S_F^2 (1 + \frac{1}{\gamma}) \beta^2 (q - 1) \sum_{s=s_t q+1}^{t-1} \alpha^2 \sum_{n=1}^{N} \mathbb{E} \| u_s^n - \bar{u}_s \|^2 \tag{42}
$$

where (a) is due to (17). Then we have

$$
\sum_{n=1}^{N} \mathbb{E} \| u_{t+1}^n - \bar{u}_{t+1} \|^2 = [(1 - \beta)^2 (1 + \gamma) + 4 S_F^2 (1 + \frac{1}{\gamma}) \alpha^2] \sum_{n=1}^{N} \mathbb{E} \| u_t^n - \bar{u}_t \|^2
$$

$$
+ 4 N S_F^2 (1 + \frac{1}{\gamma}) \alpha^2 \mathbb{E} \| \bar{u}_t \|^2 + 52 N L_f^2 L_g^2 (1 + \frac{1}{\gamma}) \beta^2
$$

$$
+ 24 S_F^2 (1 + \frac{1}{\gamma}) \beta^2 (q - 1) \sum_{s=s_t q+1}^{t-1} \alpha^2 \sum_{n=1}^{N} \mathbb{E} \| u_s^n - \bar{u}_s \|^2 \tag{43}
$$

Set $\gamma = \frac{1}{q}$ and $\alpha \leq \frac{1}{12 S_F q}$, and given that $\beta \in (0, 1)$,

$$
(1 - \beta)^2 (1 + \gamma) + 4 S_F^2 (1 + \frac{1}{\gamma}) \alpha^2 \leq 1 + \frac{1}{q} + 4 S_F^2 (1 + q) \alpha^2 \leq 1 + \frac{1}{q} + \frac{q+1}{36 q^2} \leq 1 + \frac{19}{18 q} \tag{44}
$$

Putting the (44) in (43), and considering $\gamma = \frac{1}{q}$ and $\alpha \le \frac{1}{12S_F q}$, $\beta = c\alpha^2$, we have

$$\sum_{n=1}^{N} \mathbb{E}\|u_{t+1}^n - \bar{u}_{t+1}\|^2$$

$$\le (1 + \frac{19}{18q}) \sum_{n=1}^{N} \mathbb{E}\|u_t^n - \bar{u}_t\|^2 + 4NS_F^2(1 + \frac{1}{\gamma})\alpha^2 \mathbb{E}\|\bar{u}_t\|^2$$

$$+ 52NL_f^2 L_g^2(1 + \frac{1}{\gamma})\beta^2 + 24S_F^2(1 + \frac{1}{\gamma})\beta^2(q-1) \sum_{s=s_t q+1}^{t-1} \alpha^2 \sum_{n=1}^{N} \mathbb{E}\|u_s^n - \bar{u}_s\|^2$$

$$\le (1 + \frac{19}{18q}) \sum_{n=1}^{N} \mathbb{E}\|u_t^n - \bar{u}_t\|^2 + \frac{2NS_F}{3}\alpha \mathbb{E}\|\bar{u}_t\|^2$$

$$+ \frac{26NL_f^2 L_g^2 c^2}{3S_F}\alpha^3 + 24S_F^2 q^2 c^2 \alpha^4 \sum_{s=s_t q+1}^{t-1} \alpha^2 \sum_{n=1}^{N} \mathbb{E}\|u_s^n - \bar{u}_s\|^2 \qquad (45)$$

We know that when $\mod(t, q) = 0$ (i.e. $t = s_t q$), $\sum_{n=1}^{N} \|u_t^n - \bar{u}_t\|^2 = 0$

$$\sum_{n=1}^{N} \mathbb{E}\|u_{t+1}^n - \bar{u}_{t+1}\|^2$$

$$\le \frac{2NS_F}{3} \sum_{s=s_t q}^{t} (1 + \frac{19}{18q})^{t-s} \alpha \mathbb{E}\|\bar{u}_s\|^2 + \frac{26NL_f^2 L_g^2 c^2}{3S_F} \sum_{s=s_t q}^{t} (1 + \frac{19}{18q})^{t-s} \alpha^3$$

$$+ 24S_F^2 q^2 c^2 \sum_{s=s_t}^{t} (1 + \frac{19}{18q})^{t-s} \alpha^4 \sum_{\bar{s}=s_t q}^{s} \alpha^2 \sum_{n=1}^{N} \mathbb{E}\|u_{\bar{s}}^n - \bar{u}_{\bar{s}}\|^2$$

$$\le \frac{2NS_F}{3} \sum_{s=s_t q}^{t} (1 + \frac{19}{18q})^{q} \alpha \mathbb{E}\|\bar{u}_s\|^2 + \frac{26NL_f^2 L_g^2 c^2}{3S_F} \sum_{s=s_t q}^{t} \left(1 + \frac{19}{18q}\right)^{q} \alpha^3$$

$$+ 24S_F^2 q^3 c^2 (\frac{1}{12S_F q})^5 (1 + \frac{19}{18q})^{q} \sum_{s=s_t q}^{t} \alpha \sum_{n=1}^{N} \mathbb{E}\|u_s^n - \bar{u}_s\|^2$$

$$\le 2NS_F \sum_{s=s_t q}^{t} \alpha \mathbb{E}\|\bar{u}_s\|^2 + \frac{26NL_f^2 L_g^2 c^2}{S_F} \sum_{s=s_t q}^{t} \alpha^3 + 72S_F^2 q^3 c^2 (\frac{1}{12S_F q})^5 \sum_{s=s_t q}^{t} \alpha \sum_{n=1}^{N} \mathbb{E}\|u_s^n - \bar{u}_s\|^2$$

where the third inequality is due to $(1 + 19/18q)^q \le e^{19/18} \le 3$. Multiplying $\alpha$ on both side and summing over $[s_t q, \bar{s})$ in one inner loop, where $\bar{s} = (s_t + 1)q$ we have

$$\sum_{t=s_t q}^{\bar{s}} \alpha \sum_{n=1}^{N} \mathbb{E}\|u_{t+1}^n - \bar{u}_{t+1}\|^2 \le 2NS_F \sum_{t=s_t q}^{\bar{s}} \alpha \sum_{s=s_t q}^{t} \alpha \mathbb{E}\|\bar{u}_s\|^2 + \frac{26NL_f^2 L_g^2 c^2}{S_F} \sum_{t=s_t q}^{\bar{s}} \alpha \sum_{s=s_t q}^{t} \alpha^3$$

$$+ 72S_F^2 q^3 c^2 (\frac{1}{12S_F q})^5 \sum_{t=s_t q}^{\bar{s}} \alpha \sum_{s=s_t q}^{t} \alpha \sum_{n=1}^{N} \mathbb{E}\|u_s^n - \bar{u}_s\|^2$$

$$\le 2NS_F (\sum_{t=s_t q}^{\bar{s}} \alpha) \sum_{t=s_t q}^{\bar{s}} \alpha \mathbb{E}\|\bar{u}_t\|^2 + \frac{26NL_f^2 L_g^2 c^2}{S_F} (\sum_{t=s_t q}^{\bar{s}} \alpha) \sum_{t=s_t q}^{\bar{s}} \alpha^3$$

$$+ 72S_F^2 q^3 c^2 (\frac{1}{12S_F q})^5 (\sum_{t=s_t}^{\bar{s}} \alpha) \sum_{t=s_t q}^{\bar{s}} \alpha \sum_{n=1}^{N} \mathbb{E}\|u_t^n - \bar{u}_t\|^2$$

$$\le \frac{N}{6} \sum_{t=s_t q}^{\bar{s}} \alpha \mathbb{E}\|\bar{u}_t\|^2 + \frac{13NL_f^2 L_g^2 c^2}{6S_F^2} \sum_{t=s_t q}^{\bar{s}} \alpha^3 + 72S_F^2 q^4 c^2 (\frac{1}{12S_F q})^6 \sum_{t=s_t q}^{\bar{s}+1} \alpha \sum_{n=1}^{N} \mathbb{E}\|u_t^n - \bar{u}_t\|^2$$

Rearranging the terms, we get,

$$[1 - 72S_F^2 q^4 c^2 (\frac{1}{12S_F q})^6] \sum_{t=s_t q+1}^{\bar{s}+1} \alpha \sum_{n=1}^{N} \mathbb{E}\|u_t^n - \bar{u}_t\|^2 \leq \frac{N}{6} \sum_{t=s_t q}^{\bar{s}} \alpha \mathbb{E}\|\bar{u}_t\|^2 + \frac{13NL_f^2 L_g^2 c^2}{6S_F^2} \sum_{t=s_t q}^{\bar{s}} \alpha^3$$

Given that $c = \frac{30S_F^2}{bN}$, and $(1 - 72S_F^2 q^4 c^2 (\frac{1}{12S_F q})^6)/2 \geq \frac{101}{240}$. By multiply $\frac{1}{2N}$ on both size and summing t from 1 to $T$, we have

$$\frac{101}{240N} \sum_{t=1}^{T} \alpha \sum_{n=1}^{N} \mathbb{E}\|u_t^n - \bar{u}_t\|^2 \leq \frac{1}{12} \sum_{t=1}^{T} \alpha \mathbb{E}\|\bar{u}_t\|^2 + \frac{13L_f^2 L_g^2 c^2}{12S_F^2} T\alpha^3 \tag{46}$$

$\square$

## D.2 Proof of Theorem 4.7

In this section, we show the Proof of Theorem 4.7.

*Proof.* Set $\alpha = \frac{1}{12qS_F}$, $\quad \beta = c \cdot \alpha^2$, $c = \frac{30S_F^2}{bN}$. Recall Lemma D.2, we have

$$\mathbb{E}\|\bar{u}_{t+1} - \nabla \hat{F}(x_t)\|^2$$

$$\leq (1-\beta)^2 \mathbb{E}\|\bar{u}_t - \nabla \hat{F}(x_{t-1})\|^2 + \frac{8(1-\beta)^2 S_F^2}{N^2 b} \frac{q-1}{q} \sum_{n=1}^{N} \alpha^2 \mathbb{E}\|u_t^n - \bar{u}_t\|^2$$

$$+ \frac{4(1-\beta)^2 S_F^2 \alpha^2}{Nb} \mathbb{E}\|\bar{u}_t\|^2 + \frac{2\beta^2 L_f^2}{Nb}$$

$$\leq (1-\beta) \mathbb{E}\|\bar{u}_t - \nabla \hat{F}(x_{t-1})\|^2 + \frac{8S_F^2}{N^2 b} \sum_{n=1}^{N} \alpha^2 \mathbb{E}\|u_t^n - \bar{u}_t\|^2 + \frac{4S_F^2 \alpha^2}{Nb} \mathbb{E}\|\bar{u}_t\|^2 + \frac{2\beta^2 L_f^2}{Nb} \tag{47}$$

We define the potential function as a linear combination of the objective function and the gradient estimation error:

$$\Phi_t = F(\bar{x}_t) + \frac{3\alpha}{2\beta} \|\bar{u}_{t+1} - \nabla \hat{F}(x_t)\|^2 \tag{48}$$

Therefore, we have

$$\mathbb{E}\Phi_{t+1} - \mathbb{E}\Phi_t = \mathbb{E}[F(\bar{x}_{t+1}) + \frac{3\alpha}{2\beta} \|\bar{u}_{t+2} - \nabla \hat{F}(x_{t+1})\|^2] - \mathbb{E}[F(\bar{x}_t) + \frac{3\alpha}{2\beta} \|\bar{u}_{t+1} - \nabla \hat{F}(x_t)\|^2]$$

$$\leq -(\frac{\alpha}{2} - \frac{\alpha^2 S_F}{2}) \mathbb{E}\|\bar{u}_{t+1}\|^2 - \frac{\alpha}{2} \mathbb{E}\|\nabla F(\bar{x}_t)\|^2 + \frac{3\alpha S_F^2}{2N} \mathbb{E}\|x_t^n - \bar{x}_t\|^2 + \frac{3\alpha L_g^2 S_f^2 \sigma_g^2}{2m}$$

$$+ \frac{3\alpha}{2} \mathbb{E}\|\frac{1}{N} \sum_{n=1}^{N} \nabla \hat{F}^n(x_t) - \bar{u}_{t+1}\|^2 - \frac{3\alpha}{2} \mathbb{E}\|\bar{u}_{t+1} - \frac{1}{N} \sum_{n=1}^{N} \nabla \hat{F}^n(x_t^n)\|^2$$

$$+ \frac{12\alpha S_F^2}{N^2 b\beta} \sum_{n=1}^{N} \alpha^2 \mathbb{E}\|u_{t+1}^n - \bar{u}_{t+1}\|^2 + \frac{6S_F^2 \alpha^3}{Nb\beta} \mathbb{E}\|\bar{u}_{t+1}\|^2 + \frac{3\alpha\beta L_f^2}{Nb} \tag{49}$$

Rearranging (49) and taking the telescoping sum over t in $[s_t q, \bar{s}]$ in one inner loop, where $\bar{s} = (s_t + 1)q$, we have

$$
\sum_{t=s_t q}^{\bar{s}-1} \frac{\alpha}{2} \mathbb{E}\|\nabla F(\bar{x}_t)\|^2
$$

$$
\leq \sum_{t=s_t q}^{\bar{s}-1} [\mathbb{E}\Phi_t - \mathbb{E}\Phi_{t+1}] - (\frac{\alpha}{2} - \frac{\alpha^2 S_F}{2} - \frac{6 S_F^2 \alpha^3}{Nb\beta}) \sum_{t=s_t q}^{\bar{s}-1} \mathbb{E}\|\bar{u}_{t+1}\|^2 + \frac{3\alpha S_F^2}{2N} \sum_{t=s_t q}^{\bar{s}-1} \mathbb{E}\|x_t^n - \bar{x}_t\|^2
$$

$$
+ \sum_{t=s_t q}^{\bar{s}-1} \frac{3\alpha L_g^2 S_f^2 \sigma_g^2}{2m} + \frac{12\alpha S_F^2}{N^2 b\beta} \sum_{t=s_t q}^{\bar{s}-1} \sum_{n=1}^{N} \alpha^2 \mathbb{E}\|u_{t+1}^n - \bar{u}_{t+1}\|^2 + \sum_{t=s_t q}^{\bar{s}-1} \frac{3\alpha\beta L_f^2}{Nb}
$$

$$
\leq \sum_{t=s_t q}^{\bar{s}-1} [\mathbb{E}\Phi_t - \mathbb{E}\Phi_{t+1}] - (\frac{\alpha}{2} - \frac{\alpha^2 S_F}{2} - \frac{6 S_F^2 \alpha^3}{Nb\beta}) \sum_{t=s_t q}^{\bar{s}-1} \mathbb{E}\|\bar{u}_{t+1}\|^2
$$

$$
+ \frac{3\alpha S_F^2}{2N} \sum_{t=s_t q}^{\bar{s}-1} \mathbb{E}[(q-1)\alpha^2 \sum_{s=s_t q}^{t} \sum_{n=1}^{N} \|u_s^n - \bar{u}_s\|^2] + \sum_{t=s_t q}^{\bar{s}-1} \frac{3\alpha L_g^2 S_f^2 \sigma_g^2}{2m}
$$

$$
+ \frac{12\alpha S_F^2}{N^2 b\beta} \sum_{t=s_t q}^{\bar{s}-1} \sum_{n=1}^{N} \alpha^2 \mathbb{E}\|u_{t+1}^n - \bar{u}_{t+1}\|^2 + \sum_{t=s_t q}^{\bar{s}-1} \frac{3\alpha\beta L_f^2}{Nb} \tag{50}
$$

where the last inequality uses (17). Furthermore,

$$
\sum_{t=s_t q}^{\bar{s}-1} \frac{\alpha}{2} \|\nabla F(\bar{x}_t)\|^2 \leq \sum_{t=s_t q}^{\bar{s}-1} [\mathbb{E}\Phi_t - \mathbb{E}\Phi_{t+1}] - (\frac{\alpha}{2} - \frac{\alpha^2 S_F}{2} - \frac{6 S_F^2 \alpha^3}{Nb\beta}) \sum_{t=s_t q}^{\bar{s}-1} \mathbb{E}\|\bar{u}_{t+1}\|^2
$$

$$
+ \sum_{t=s_t q}^{\bar{s}-1} \frac{3\alpha L_g^2 S_f^2 \sigma_g^2}{2m} + \frac{3 S_F^2 \alpha}{2N} [(q-1) \times q \times \frac{1}{12 S_F q} \times \frac{1}{12 S_F q} \sum_{t=s_t q}^{\bar{s}-1} \sum_{n=1}^{N} \|u_{t+1}^n - \bar{u}_{t+1}\|^2]
$$

$$
+ \sum_{t=s_t q}^{\bar{s}-1} [\frac{12 S_F^2 \alpha^3}{N^2 b\beta} \sum_{n=1}^{N} \mathbb{E}\|u_{t+1}^n - \bar{u}_{t+1}\|^2 + \frac{3\alpha\beta L_f^2}{Nb}]
$$

$$
\leq \sum_{t=s_t q}^{\bar{s}-1} \mathbb{E}[\Phi_t - \Phi_{t+1}] - (\frac{\alpha}{2} - \frac{\alpha^2 S_F}{2} - \frac{6 S_F^2 \alpha^3}{Nb\beta}) \sum_{t=s_t q}^{\bar{s}-1} \mathbb{E}\|\bar{u}_{t+1}\|^2 + \sum_{t=s_t q}^{\bar{s}-1} \frac{3\alpha L_g^2 S_f^2 \sigma_g^2}{2m}
$$

$$
+ \frac{\alpha}{96N} \sum_{t=s_t q}^{\bar{s}-1} \sum_{n=1}^{N} \|u_{t+1}^n - \bar{u}_{t+1}\|^2 + \sum_{t=s_t q}^{\bar{s}-1} [\frac{12 S_F^2 \alpha^3}{N^2 b\beta} \sum_{n=1}^{N} \mathbb{E}\|u_{t+1}^n - \bar{u}_{t+1}\|^2 + \frac{3\alpha\beta L_f^2}{Nb}]
$$

$$
\leq \sum_{t=s_t q}^{\bar{s}-1} \mathbb{E}[\Phi_t - \Phi_{t+1}] - (\frac{\alpha}{2} - \frac{\alpha^2 S_F}{2} - \frac{6 S_F^2 \alpha}{Nbc}) \sum_{t=s_t q}^{\bar{s}-1} \|\bar{u}_{t+1}\|^2 + \sum_{t=s_t q}^{\bar{s}-1} \frac{3\alpha L_g^2 S_f^2 \sigma_g^2}{2m}
$$

$$
+ (\frac{1}{96N} + \frac{12 S_F^2}{N^2 bc})\alpha[\sum_{t=s_t q}^{\bar{s}-1} \sum_{n=1}^{N} \|u_{s+1}^n - \bar{u}_{s+1}\|^2] + \sum_{t=s_t q}^{\bar{s}-1} \frac{3\beta\alpha L_f^2}{Nb}
$$

$$
= \sum_{t=s_t q}^{\bar{s}-1} \mathbb{E}[\Phi_t - \Phi_{t+1}] - (\frac{3\alpha}{10} - \frac{\alpha^2 S_F}{2}) \sum_{t=s_t q}^{\bar{s}-1} \mathbb{E}\|\bar{u}_{t+1}\|^2 + \frac{101\alpha}{240N} \sum_{t=s_t q}^{\bar{s}-1} \sum_{n=1}^{N} \|u_s^n - \bar{u}_s\|^2
$$

$$
+ \sum_{t=s_t q}^{\bar{s}-1} \frac{3\alpha L_g^2 S_f^2 \sigma_g^2}{2m} + \sum_{t=s_t q}^{\bar{s}-1} \frac{3\beta\alpha L_f^2}{Nb}
$$

where the last equality holds by to $c = \frac{30S_F^2}{bN}$. Therefore, by summing t from 0 to $T$, we have

$$\sum_{t=0}^{T-1} \frac{\alpha}{2} \|\nabla F(\bar{x}_t)\|^2$$

$$\leq \mathbb{E}[\Phi_0 - \Phi_T] - (\frac{3\alpha}{10} - \frac{\alpha^2 S_F}{2}) \sum_{t=0}^{T-1} \mathbb{E}\|\bar{u}_{t+1}\|^2 + \frac{101\alpha}{240N} \sum_{t=0}^{T-1} \sum_{n=1}^{N} \|u_s^n - \bar{u}_s\|^2$$

$$+ \frac{3\alpha L_g^2 S_f^2 \sigma_g^2 T}{2m} + \frac{3\beta \alpha L_f^2 T}{Nb}$$

$$\overset{(a)}{\leq} \mathbb{E}[\Phi_0 - \Phi_T] - (\frac{3\alpha}{10} - \frac{\alpha}{12} - \frac{\alpha^2 S_F}{2}) \sum_{t=0}^{T-1} \mathbb{E}\|\bar{u}_{t+1}\|^2 + \frac{13L_f^2 L_g^2 c^2}{12S_F^2} T\alpha^3$$

$$+ \frac{3\alpha L_g^2 S_f^2 \sigma_g^2 T}{2m} + \frac{3\beta \alpha L_f^2 T}{Nb}$$

$$\leq \mathbb{E}[\Phi_0 - \Phi_T] + \frac{13L_f^2 L_g^2 c^2}{12S_F^2} T\alpha^3 + \frac{3\alpha L_g^2 S_f^2 \sigma_g^2 T}{2m} + \frac{3\beta \alpha L_f^2 T}{Nb}$$

$$\leq [F(\bar{x}_0) - F(\bar{x}_T)] + [\frac{3\alpha}{2\beta} \mathbb{E}\|\bar{u}_1 - \frac{1}{N} \sum_{n=1}^{N} \nabla \hat{F}^n(x_0^n)\|^2 + \frac{13L_f^2 L_g^2 c^2}{12S_F^2} T\alpha^3$$

$$+ \frac{3\alpha L_g^2 S_f^2 \sigma_g^2 T}{2m} + \frac{3\beta \alpha L_f^2 T}{Nb}$$

where (a) holds due to D.3. Therefore,

$$\frac{1}{T} \sum_{t=0}^{T-1} \|\nabla F(\bar{x}_t)\|^2 \leq \frac{2[F(\bar{x}_0) - F(\bar{x}_T)]}{T\alpha} + \frac{3L_f^2 L_g^2}{\beta BNT} + \frac{13L_f^2 L_g^2 c^2}{6S_F^2} \alpha^2 + \frac{3L_g^2 S_f^2 \sigma_g^2}{m} + \frac{6\beta L_f^2}{Nb}$$

let b as $O(1)(b \geq 1)$, and choose $q = (T/N^2)^{1/3}$. Therefore, $\alpha = \frac{1}{12qS_F} = \frac{N^{2/3}}{12S_F T^{1/3}}$, since $c = \frac{30S_F^2}{bN}$, we have $\beta = c\alpha^2 = \frac{5N^{1/3}}{24T^{2/3}b}$. And let $B = \frac{T^{1/3}}{N^{2/3}}$

Therefore, we have

$$\frac{1}{T} \sum_{t=0}^{T-1} \|\nabla F(\bar{x}_t)\|^2$$

$$\leq \frac{24S_F[F(\bar{x}_0) - F(\bar{x}_*)]}{(NT)^{2/3}} + \frac{72L_f^2 L_g^2 b}{5(NT)^{2/3}} + \frac{325L_f^2 L_g^2}{24b^2(TN)^{2/3}} + \frac{3L_g^2 S_f^2 \sigma_g^2}{m} + \frac{5L_f^2}{4b^2(NT)^{2/3}} \quad (51)$$

To let the right hand less than $\varepsilon^2$ when $m = \varepsilon^{-2}$ and $b = O(1)$, we get $T = O(N^{-1}\varepsilon^{-3})$ and $\frac{T}{q} = (NT)^{2/3} = \varepsilon^{-2}$.

$\square$