# OpenReview forum: "Federated Conditional Stochastic Optimization"
_NeurIPS.cc/2023/Conference — NeurIPS 2023 poster_

### Official Review · Reviewer_pkoz · 2023-07-02

**Soundness:** 3 good
**Presentation:** 3 good
**Contribution:** 3 good
**Rating:** 7
**Confidence:** 3

**Summary:**

This work proposes the first provably efficient federated conditional stochastic optimization solutions with a biased gradient estimator, including 3 variants: FCSG, FCSG-M (with momentum), and Acc-FCSG-M (with a variance reduction technique). The sample complexity and communication complexity are analyzed for all 3 algorithms. Notably, Acc-FCSG-M achieves the lower bound of sample complexity of the single-machine counterpart. Additionally, the paper demonstrates the empirical performance of these algorithms on robust logistic regression and MAML tasks.

**Strengths:**

This research represents a pioneering investigation into the nonconvex conditional stochastic optimization (CSO) within the federated learning framework. It presents three algorithms and conducts a comprehensive analysis of their sample and communication complexities. Notably, the Acc-FCSG-M matches the lower bound of sample complexity of single-machine CSO.

**Weaknesses:**

There are two areas where the paper could be enhanced. Firstly, the introduction should provide a stronger motivation for federated conditional stochastic optimization (CSO). Currently, the paper separately emphasizes the significance of CSO and federated learning, but fails to incorporate specific real-world problems that would necessitate the use of federated CSO. Secondly, the paper should emphasize its technical contribution by briefly highlighting the unique aspects in the proof that are absent in prior works.

**Questions:**

I don't have any additional suggestions besides the ones mentioned in the aforementioned weaknesses.

**Limitations:**

The authors have adequately addressed the limitations of the paper.

---

> ### Author Rebuttal · Authors · 2023-08-09
>
> I really appreciate your time and comments. I would follow your suggestion and add more details in final version.
>
> The technical contribution in the proof that are absent in prior works is listed here:
>
> First, we are the first work to do federated conditional stochastic optimization and we not only need to consider the complicated structure of the CSO problem but also need to handle the local update and global update separately.  Second, in the case of Acc-FCSG-M, a direct combination of variance reduction mechanisms like STORM with FL conditional stochastic optimization cannot work because the specially calibrated step sizes in STORM. We use different theoretical analyses and our approach involves a more lenient step size condition. At the same time, our results is tighter and does not include logarithmic terms, which are present in the outcomes of STORM-like methodologies [1] [2] .
>
> [1] Cutkosky, Ashok, and Francesco Orabona. "Momentum-based variance reduction in non-convex sgd." Advances in neural information processing systems 32 (2019).
>
> [2] Khanduri, Prashant, et al. "Stem: A stochastic two-sided momentum algorithm achieving near-optimal sample and communication complexities for federated learning." Advances in Neural Information Processing Systems 34 (2021): 6050-6061. https://proceedings.neurips.cc/paper/2021/file/3016a447172f3045b65f5fc83e04b554-Paper.pdf

---

> ### Author Response · Authors · 2023-08-17
> **Extra experiments in FL AUPRC maximization**
>
> > Q1." ... the introduction should provide a stronger motivation for federated conditional stochastic optimization (CSO) ... "
>
> We add extra experiments about Federated AUPRC maximization here. AUROC maximization in FL have been studied in [2][3] and AUPRC maximization is also used to solve the imbalanced classification. Existing AUPRC algorithms maintain an inner state for each data point. For the large-scale distributed data over multiple clients, algorithms for online  AUPRC maximization in FL is necessary.
>
> Following the eq (2) in [1], the online AUPRC maximization could be reformulated as:
>
> $ \begin{align}
> \min P = - \mathbb{E}\_{\mathbf{x_i} \sim  \mathcal{D}_{+}} \frac{\mathbb{E}\_{\mathbf{x_s} \sim  \mathcal{D}} \mathbf{I}\left(y_s=1\right)  \ell \left(\mathbf{w}; \mathbf{x}_s, \mathbf{x}_i \right)}{\mathbb{E}\_{\mathbf{x_s} \sim \mathcal{D}} \quad \ell\left(\mathbf{w}; \mathbf{x}_s, \mathbf{x}_i\right)}
> \end{align}$
>
> Following the eq(4) in [1], FL objective function is listed below:
> $\min \frac{1}{K} \sum P\_k (\mathbf{w}) = \min \frac{1}{K} \sum\mathbb{E}\_{\mathbf{x_i} \sim  \mathcal{D}^k\_{+}} [f^k(g^k_{\mathbf{x}_i} (\mathbf{w}))]$
>
>
> It is a two-level problem and $x_i$ is sampled from positive datasets and $x_s$ is sampled to estimate the score rank of $x_i$ . The inner objective depends on both $x_i$ and $x_s$. Since federated AUPRC is a special example of a federated CSO, our algorithms could be directly applied to it.
>
> We choose MNIST dataset and CIFAR-10 datasets. Given that we cannot upload the images, we make a table to show our results.
>
> As AUROC maximization in federated settings has been demonstrated in existing works, e.g., [2, 3]. we use CODA+ in [3] as a baseline. In addition, we also use FedAvg with cross-entropy loss as a baseline. Since we cannot submit the external link, we will release related code in the final version.
>
> Since AUPRC is used for binary classification, the first half of the classes in the MNIST and CIFAR10 datasets are designated to be a negative class, and the rest half of the classes are considered to be the positive class. Then, we remove 80% of the positive examples in the training set to make it imbalanced, while keeping the test set unchanged. The results show that our algorithms could be used to solve the online AUPRC maximization in FL and it largely improves the model's performance.
>
> | Datasets |  SGD + CE |  CODA+ | FCSG | FCSG-M | Acc-FCSG-M |
> | ---------- | ------------- | --------- | ------- | ---------- | ---------------- |
> | MNIST    |     0.9357    | 0.9733   |0.9868 | 0.9878     |   0.9879           |
> | CIFAR-10| 0.5059       | 0.6039  |  0.7130 |  0.7157    |  0.7184           |
>
>
> [1] Qi, Q., Luo, Y., Xu, Z., Ji, S., and Yang, T. Stochastic optimization of areas under precision-recall curves with provable convergence. Advances in Neural Information Processing Systems, 2021. https://arxiv.org/pdf/2104.08736.pdf
>
> [2] Guo, Zhishuai, Mingrui Liu, Zhuoning Yuan, Li Shen, Wei Liu, and Tianbao Yang. "Communication-efficient distributed stochastic auc maximization with deep neural networks." In International conference on machine learning, pp. 3864-3874. PMLR, 2020.
>
> [3] Z Yuan, Z Guo, Y Xu, Y Ying, T Yang. Federated Deep AUC Maximization with a Constant Communication Complexity, In International conference on machine learning, 2021.

---

> > ### Comment · Reviewer_pkoz · 2023-08-19
> > **Thank Authors For The Detailed Response**
> >
> > Thank the authors for clarifying the technical challenges and presenting additional detailed experiment results.
> > The efforts have effectively resolved my concerns.

---

> > > ### Author Response · Authors · 2023-08-19
> > >
> > > We are deeply thankful for your appreciation and valuable comments

---

### Official Review · Reviewer_jBXN · 2023-07-04

**Soundness:** 3 good
**Presentation:** 3 good
**Contribution:** 2 fair
**Rating:** 6
**Confidence:** 3

**Summary:**

This paper explores different algorithms for optimization problems in the FL setting with 2 layers of stochasticity, where the inner stochasticity is dependent on the outer layer. An interesting application for these algorithms is the MAML problem, where we aim to find a global initialization over clients that carry data from different tasks. Inspired by recent work in stochastic conditional optimization, the authors propose a simple algorithm that uses biased gradient estimates and also propose momentum and variance reduction extensions.
For all 3 algorithms, both a convergence guarantee, communication complexity and sample complexity is given.
Finally empirical evidence is provided on invariant logistic regression and the MAML problem on the Omniglot dataset. They compare against Stochastic Compositional optimization algorithms on the MAML problem.

**Strengths:**

- The paper is understandable for someone who is not familiar with stochastic conditional optimization problems
- Conditional stochastic optimization in FL is a novel problem formulation that is relevant, especially when viewed from the perspective of MAML, as non-iid-ness of client distributions is still one of the most important challenges to tackle in FL.
- The convergence analysis is extensive

**Weaknesses:**

- The proposed method is only evaluated on problems where it is assumes multiple tasks can be sampled at clients instead of one task being assumed for each client. While tackling this problem setting can be a valid contribution, it is questionable to me how realistic this assumed setting is. The authors motivate this assumption around line 258 by saying that 'we can evenly distribute the tasks along the worker nodes', but this is not how the FL setting works in practice. If the proposed algorithm also shows strong performance on the ‘one task per client’ problem, which is a common way of tackling the non-iid-ness issue in FL, this would make for stronger contribution, but experiments on this problem setting are missing.
- The proposed algorithms seem to assume that all clients participate in each communication round, which is impractical and unscalable in the cross-device FL setting where a high number of clients is typical and the presence of stragglers is probable.
- The paper contains some grammar mistakes
- Adding some naive baselines on the MAML problem such as e.g [1][2] and FedAvg would give a better insight in the contribution of the proposed method.
- No results are given of the eventual ‘converged’ accuracy/loss of the algorithms and the baselines, merely convergence results are shown. Therefore the claim ‘our algorithms outperform baselines by a large margin’ is a little premature in my opinion, especially looking at the convergence results in figure 3.
- Some interesting questions are not answered by the empirical results, such as:
    - How do the amount of local updates influence performance?
    - How do dataset sizes of clients influence performance?

[1] Fallah, A., Mokhtari, A., & Ozdaglar, A. (2020). Personalized federated learning: A meta-learning approach. arXiv preprint arXiv:2002.07948.

[2] Jiang, Y., Konečný, J., Rush, K., & Kannan, S. (2019). Improving federated learning personalization via model agnostic meta learning. arXiv preprint arXiv:1909.12488.

**Questions:**

- Does the algorithm allow for partial participation?
- How does this algorithm perform in the one-task-per-client setting, as [1] and [2] are focussing on?
- In section 5.2, do you only consider $q=1$, concluding from line 274?
- Are the experimental results concluded from multiple runs from different initializations?

**Limitations:**

- As stated before, it is not clear whether the algorithm also applies to partial participation settings. If not, then this is a limitation in the cross-device FL setting

---

> ### Author Rebuttal · Authors · 2023-08-09
>
> Thanks a lot for your time and valuable review.
>
> > Q1. "The proposed method is only evaluated on problems where it is assumes multiple tasks can be sampled at clients instead of one task being assumed for each client. "
>
> Our paper focus on conditional stochastic optimization and FL MAML is an example while Personalized federated learning is different.  In the related work, we mention some work that applies the personalization problem in federated learning in section 2.2. “Stochastic Composition Optimization” and it could be regarded as federated stochastic composition optimization.
>
> We would like to reiterate one key difference between Stochastic Composition Optimization (SCO) and Conditional Stochastic Optimization (CSO): in the SCO the sampling from the inner randomness and the outer randomness are $\mathbf{independent}$, while in Problem CSO, they are not. Task sampling is the main challenge in MAML and that is why we need to study Federated Conditional Stochastic Optimization. Some works [2][3] have studied stochastic composition optimization and propose related theoretical analysis. It is not in the scope of our work.
>
>
> > Q2. "Does the algorithm allow for partial participation?"
>
> By adding the clients sampling, our algorithms can consider partial participation. The theoretical analysis of  partial participation will be consider in future work.
>
>
> > Q2. "How does this algorithm perform in the one-task-per-client setting, as [1] and [2] are focussing on?"
>
> Seen as Q1.
>
> > Q3.  "In section 5.2, do you only consider $q = 1$, concluding from line 274?
>
> No. It is completely different. q is the local training steps in federated learning. This one step means an update step in the inner layer in the MAML task where we follow the setting in [4].
>
> > Q4. "Are the experimental results concluded from multiple runs from different initialization?"
>
> I run each test 3 times and report the average results. We will update the image with the std shadow in the final version. Thanks for your mentions.
>
>
>
>
> [1] Fallah, A., Mokhtari, A., & Ozdaglar, A. (2020). Personalized federated learning: A meta-learning approach. arXiv preprint arXiv:2002.07948.
>
> [2] Gao, H., Li, J., and Huang, H. On the convergence of local stochastic compositional gradient descent with momentum. In International Conference on Machine Learning, PMLR, 2022
>
> [3] Huang, F., Li, J., and Huang, H. Compositional federated learning: Applications in distributionally robust averaging and meta learning. arXiv preprint arXiv:2106.11264, 2021
>
>
> [4] Chelsea Finn, Pieter Abbeel, and Sergey Levine. Model-agnostic meta-learning for fast adaptation of deep networks. In International conference on machine learning, pages 1126–1135. PMLR, 2017. https://arxiv.org/pdf/1703.03400.pdf

---

> > ### Comment · Reviewer_jBXN · 2023-08-14
> > **Response to author comments**
> >
> > Thank you for your answers.
> >
> > Regarding Q1, I understand the difference between CSO and SCO, but thank you for your additional clarification. Perhaps because I fail to see imagine relevant real world examples of problems that fall in the CSO setting for FL, I'm over-focusing on MAML, which is an example I do see the relevance for in the FL setting.
> >
> > Like reviewer pkoz, I feel like the motivation for CSO is a bit lacking and I need some more example problems to see the relevance of this method. I will stick to my score, because I think it is still a useful contribution and the paper is generally sound, but will consider to increase it if you can address this a bit better.
> >
> > Also, please have a second look at some of the grammar, e.g line 108 "... studied stochastic compositional problem ..." --> "... studied THE stochastic compositional problem ..."

---

> > > ### Author Response · Authors · 2023-08-14
> > > **Thanks for your response**
> > >
> > > We truly thank your response and appreciation.
> > >
> > > > Q1. " ... real world examples of problems ... "
> > >
> > > We do have several motivational examples for CSO in FL setting, other than MAML. For example,
> > >
> > > - Federated AUPRC Maximization. It is known that in many important applications where the data for classification is highly imbalanced, areas under the precision-recall curves (AUPRC) is a much more suitable measure than accuracy to be optimized. The necessity and importance of deploying AUROC maximization in federated settings have been demonstrated in existing works, e.g., [3, 4]. Notably, AUPRC maximization [1] could be regarded as CSO problem. It is formulated as eq. (2) and (4) in [2]. It is a two-level problem and $x_i$ is sampled from positive datasets and $x_j$ is sampled to estimate the score rank of $x_i$. The inner objective depends on both $x_i$  and $x_j$. Since federated AUPRC is a special example of federated CSO, our proposed algorithms in this paper can be directly applied to solve online AUPRC maximization in FL.
> > >
> > > - Federated Reinforcement Learning. The problem, where n agents collaboratively learn a single policy without sharing the trajectories from their respective agent-environment interaction, is considered in existing work e.g. [5]. The objective function also belongs to CSO (please see eq. (2) and (3) in [6]). Thus, our analysis also applies to this class of problems.
> > >
> > > We will add these motivational examples to our paper.
> > >
> > >
> > > > Q2. " ... some of the grammar ..."
> > >
> > > Thanks for pointing them out. We will fix these grammar mistakes in the final version.
> > >
> > >
> > >
> > > [1] Qi, Q., Luo, Y., Xu, Z., Ji, S., and Yang, T. Stochastic optimization of areas under precision-recall curves with provable convergence. Advances in Neural Information Processing Systems, 2021. https://arxiv.org/pdf/2104.08736.pdf
> > >
> > > [2] Wang, Guanghui, Ming Yang, Lijun Zhang, and Tianbao Yang. "Momentum accelerates the convergence of stochastic auprc maximization." In International Conference on Artificial Intelligence and Statistics, pp. 3753-3771. PMLR, 2022. https://proceedings.mlr.press/v151/wang22b/wang22b.pdf
> > >
> > > [3] Guo, Zhishuai, Mingrui Liu, Zhuoning Yuan, Li Shen, Wei Liu, and Tianbao Yang. "Communication-efficient distributed stochastic auc maximization with deep neural networks." In International conference on machine learning, pp. 3864-3874. PMLR, 2020.
> > >
> > > [4] Z Yuan, Z Guo, Y Xu, Y Ying, T Yang. Federated Deep AUC Maximization with a Constant Communication Complexity, In International conference on machine learning, 2021.
> > >
> > > [5] Hao Jin, Yang Peng, Wenhao Yang, Shusen Wang, Zhihua Zhang. Federated Reinforcement Learning with Environment Heterogeneity, In  International Conference on Artificial Intelligence and Statistics (AISTATS), 2022.
> > >
> > > [6] Bo Dai, Albert Shaw, Lihong Li, Lin Xiao, Niao He, Zhen Liu, Jianshu Chen, Le Song. SBEED: Convergent Reinforcement Learning with Nonlinear Function Approximation, In the 35th International Conference on Machine Learning (ICML 2018)

---

> > > ### Author Response · Authors · 2023-08-17
> > > **Extra real world example (AUPRC maximization)**
> > >
> > > We add extra real-world examples here by using our algorithm to solve Federated AUPRC maximization.
> > >
> > > Basic information about AUPRC maximization is introduced in [1].
> > > Following the eq (2) in [1], the online AUPRC maximization could be reformulated as:
> > >
> > > $ \begin{align}
> > > \min P = - \mathbb{E}\_{\mathbf{x_i} \sim  \mathcal{D}_{+}} \frac{\mathbb{E}\_{\mathbf{x_s} \sim  \mathcal{D}} \mathbf{I}\left(y_s=1\right)  \ell \left(\mathbf{w}; \mathbf{x}_s, \mathbf{x}_i \right)}{\mathbb{E}\_{\mathbf{x_s} \sim \mathcal{D}} \quad \ell\left(\mathbf{w}; \mathbf{x}_s, \mathbf{x}_i\right)}
> > > \end{align}$
> > >
> > > Following the eq(4) in [1], FL objective function is listed below:
> > > $\min \frac{1}{K} \sum P\_k (\mathbf{w}) = \min \frac{1}{K} \sum\mathbb{E}\_{\mathbf{x_i} \sim  \mathcal{D}^k\_{+}} [f^k(g^k_{\mathbf{x}_i} (\mathbf{w}))]$
> > >
> > > It is a two-level problem and $x_i$ is sampled from positive datasets and $x_s$ is sampled to estimate the score rank of $x_i$ . The inner objective depends on both $x_i$ and $x_s$. Since federated AUPRC is a special example of a federated CSO, our algorithms could be directly applied to it.
> > >
> > > We choose MNIST dataset and CIFAR-10 datasets. Given that we cannot upload the images, we make a table to show our results.
> > >
> > > As AUROC maximization in federated settings has been demonstrated in existing works, e.g., [3, 4]. we use CODA+ in [4] as a baseline. In addition, we also use FedAvg with cross-entropy loss as a baseline. Since we cannot submit the external link, we will release related code in the final version.
> > >
> > > Since AUPRC is used for binary classification, the first half of the classes in the MNIST and CIFAR10 datasets are designated to be a negative class, and the rest half of the classes are considered to be the positive class. Then, we remove 80% of the positive examples in the training set to make it imbalanced, while keeping the test set unchanged. The results show that our algorithms could be used to solve the online AUPRC maximization in FL and it largely improves the model's performance.
> > >
> > > | Datasets |  SGD + CE |  CODA+ | FCSG | FCSG-M | Acc-FCSG-M |
> > > | ---------- | ------------- | --------- | ------- | ---------- | ---------------- |
> > > | MNIST    |     0.9357    | 0.9733   |0.9868 | 0.9878     |   0.9879           |
> > > | CIFAR-10| 0.5059       | 0.6039  |  0.7130 |  0.7157    |  0.7184           |
> > >
> > > In the final version, we will plot the test AP vs training iteration  to show the convergence rate for different algorithms.

---

> > > > ### Comment · Reviewer_jBXN · 2023-08-17
> > > > **Response to author comments**
> > > >
> > > > I thank the authors for their additional explanations and work they put in coming up with additional experiments.
> > > >
> > > > Because the motivation for the method improves with these experiments and examples added to the paper, I'm willing to raise my score.

---

> > > > > ### Author Response · Authors · 2023-08-17
> > > > >
> > > > > Thank you very much for your constructive comments and recognition of our work.

---

### Official Review · Reviewer_tZ8Z · 2023-07-05

**Soundness:** 3 good
**Presentation:** 3 good
**Contribution:** 2 fair
**Rating:** 5
**Confidence:** 4

**Summary:**

In this paper, the authors delve into the complexities of nonconvex federated conditional stochastic optimization problems. A suite of algorithms, ranging from the simplest form to accelerated versions that meet the ideal lower bounds, is presented. The study also investigates the sample and communication complexities associated with these algorithms, they also provided numerical experiments to rationalize the outperformance of proposed algorithms.

**Strengths:**

1. The paper combines both conditional stochastic optimization and federated learning.
2. The proposed accelerated algorithm matches the lower bound.
3. The writing and flow of the paper are good in general.

**Weaknesses:**

1. A bunch of assumptions are imposed here for the theoretical analysis, like Lipschitz continuity, which remedies a bunch of common issues like heterogeneity and client drift. Also it seems like the experiments you consider may not fit with your settings. I believe it is better that authors can further rationalize these assumptions.
2. As far as I can see, your paper borrows a lot of settings in the previous conditional stochastic optimization literature, which suggests that the strong assumptions you impose may be unavoidable to some extent. But with that, my second issue comes that your work seems to be just a simple extension of existing works in the centralized setting, the algorithm is obvious in my opinion; and the analysis, with the success in the centralized literature, is something that as expected I think. So the current result looks not that exciting to me in terms of significance. Could you please elaborate more on the nontrivial part in the analysis. Thank you.

Some typos:
1. Several big-O notations are missing, e.g., Line 170, 172, 191, 193, 197, 213, 214
2. Assumption 3.2, I think the rightmost term should be $L_f^2L_g^2$.

**Questions:**

See above

**Limitations:**

Not applicable.

---

> ### Author Rebuttal · Authors · 2023-08-08
>
> Thanks a lot for your time and comments.
>
> 1. "A bunch of assumptions are imposed here for the theoretical analysis, like Lipschitz continuity ... "
>
> As you mention in Q2, we follow the assumptions in the previous work and do not add any extra assumptions. For Assumption Lipschitz continuous objective (gradient bounded), it is widely used in optimization analysis in related two-level optimization. Many typical centralized two-level optimizations use this assumption [2][3][4] and FL optimization [1] [5].
>
> 2. "... your work seems to be just a simple extension of existing works in the centralized setting .. "
>
> We respectfully disagree with the statement. While FCSG is the federated counterpart of the BSGD technique introduced in [1], we take it a step further by integrating momentum technology, leading to our method named FCSG-M. We also conduct a comprehensive theoretical analysis to underscore its good performance in practical scenarios. In contrast to the centralized variance reduction approach (BSpiderBoost) outlined in [1], which mandates a substantial batch size, our proposed variance reduction method (Acc-FCSG-M) operates effectively without such a requirement.
>
> Furthermore, providing theoretical justification for these approaches is a nontrivial part. On the one hand, we not only need to consider the complicated structure of the CSO problem but also need to handle the local update and global update separately.  On the other hand, in the case of Acc-FCSG-M, a direct combination of variance reduction mechanisms like STORM [6] with FL conditional stochastic optimization is not feasible. Unlike the specially calibrated step sizes in STORM [7], our approach involves a more lenient step size condition, necessitating a distinct theoretical proof. The advantage lies in our ability to incorporate variance reduction technology. Concurrently, our results omit logarithmic terms, which are present in the outcomes of STORM-like methodologies [6] [7] (noted in a footnote on page 2 [7]).
>
>
>
>
> [1] Gao, H., Li, J., and Huang, H. On the convergence of local stochastic compositional gradient descent with momentum. In International Conference on Machine Learning, PMLR, 2022
>
> [2] Qi, Q., Luo, Y., Xu, Z., Ji, S., and Yang, T. Stochastic optimization of areas under precision-recall curves with provable convergence. Advances in Neural Information Processing Systems, 2021. https://arxiv.org/pdf/2104.08736.pdf
>
> [3] Wang, M., Fang, E. X., and Liu, H. Stochastic compositional gradient descent: algorithms for minimizing compositions of expected-value functions. Mathematical Programming, 161(1):419–449, 2017.
>
> [4] Hu, Y., Zhang, S., Chen, X., and He, N. Biased stochastic first-order methods for conditional stochastic optimization and applications in meta learning. Advances in Neural Information Processing Systems, 33:2759–2770, 2020b.
>
> [5] Huang, F., Li, J., and Huang, H. Compositional federated learning: Applications in distributionally robust averaging and meta learning. arXiv preprint arXiv:2106.11264, 2021
>
> [6] Cutkosky, Ashok, and Francesco Orabona. "Momentum-based variance reduction in non-convex sgd." Advances in neural information processing systems 32 (2019).
>
> [7] Khanduri, Prashant, et al. "Stem: A stochastic two-sided momentum algorithm achieving near-optimal sample and communication complexities for federated learning." Advances in Neural Information Processing Systems 34 (2021): 6050-6061. https://proceedings.neurips.cc/paper/2021/file/3016a447172f3045b65f5fc83e04b554-Paper.pdf

---

> ### Author Response · Authors · 2023-08-14
> **Thanks for your review**
>
> Thanks for your time and comments. Since the discussion period already began, we will appreciate it if you can check our responses and let us know if there are any further questions.

---

> ### Author Response · Authors · 2023-08-17
> **Thanks for your review**
>
> We thank you for your careful attention to our paper. As the discussion period will end in just a couple of days and we will not be able to see further comments from the reviewers, we would really appreciate it if you could review our rebuttal. If all your concerns are solved, could you please consider increasing the score? Thanks.

---

> > ### Comment · Reviewer_tZ8Z · 2023-08-17
> > **Thank you**
> >
> > Thanks to the authors for the reply.
> >
> > You mentioned "our proposed variance reduction method (Acc-FCSG-M) operates effectively without such a requirement.", but I found that your $m$ is $O(\epsilon^{-2})$, which is the same as BSpiderBoost, while $b$ is $O(1)$, which is similar to that in STORM (is that correct?). Also, you mentioned "handle the local update and global update separately", my rough understanding is that, with Lipschitz continuity of the functions (or bounded gradient norm), the heterogeneity (between gradients) should be easily bounded. So I may regard the difference you mentioned as a bit of incremental contribution.
> >
> > But I agree with the authors that the assumptions come from existing literature, also with the additional experiment presented in other discussions, I am willing to raise my score. Thank you for your efforts.

---

> > > ### Author Response · Authors · 2023-08-19
> > >
> > > $b$ is $O(1) $ and it is similar to that in STORM. But we modify the choice of step size to make it tighter without log term and it could be used in the analysis of CSO.
> > >
> > > Thanks a lot for your recognition of our work.

---

### Official Review · Reviewer_w7D4 · 2023-07-06

**Soundness:** 3 good
**Presentation:** 3 good
**Contribution:** 3 good
**Rating:** 5
**Confidence:** 2

**Summary:**

This paper proposed algorithms for federated conditional stochastic optimization problem. Noteably, the sample complexity is of $O(\epsilon^{-5})$ and the communication complexity is of $O(\epsilon^{-2})$ to obtain $\epsilon$-stationary point. This complexity matches by the single-machine algorithm BSpiderBoost with variance reduction that has been studied in Hu et. al.

**Strengths:**

The study of federated conditional stochastic optimization problem is new in literature. The proposed algorithm has sharp sample complexity rate compared with the algorithms in literature.

**Weaknesses:**

- In related work part, the reference [13] actually introduced MLMC method that has sample complexity $O(\epsilon^{-4})$, which is the optimal sample complexity. The authors omitted this fact and actually, I am wondering if it could be possible to utilize MLMC mehod to solve the federated conditional stochastic optimization problem? Then the sample complexity rate of $O(\epsilon^{-5})$ could be potentially reduced.
- In Fig. (2c) and (2d), the test loss seems does not become stable at last few iterations. The authors are suggested to increase the number of iterations to let readers see the whole trend.

**Questions:**

N/A

---

> ### Author Rebuttal · Authors · 2023-08-08
>
> Thanks for your time and valuable comments.
>
> Q1. "... the reference [13] actually introduced MLMC method that has sample complexity ..."
>
> [13] utilized the Monte Carlo method to achieve better results compared with the vanilla stochastic gradient method. In our paper, we focus on the stochastic gradient method because it is very popular in ML training. Thanks for your suggestion and we will consider it in our future work.

---

> > ### Author Response · Authors · 2023-08-11
> > **Rebuttal (continue)**
> >
> >
> > > Q2. "In Fig. (2c) and (2d), the test loss seems not to become stable at last few iterations. The authors are suggested to increase the number of iterations to let readers see the whole trend."
> >
> > Thanks for your suggestion. Since we cannot submit images, we will add them to the final version. According to the final results, our algorithms have better performance.

---

> > > ### Comment · Reviewer_w7D4 · 2023-08-19
> > > **After reading rebuttal**
> > >
> > > I thank the authors for posting their rebuttal. I will keep my score as it is.

---

> > > > ### Author Response · Authors · 2023-08-19
> > > >
> > > > Thanks a lot for your time and positive comments.

---

### Official Review · Reviewer_VagV · 2023-07-07

**Soundness:** 3 good
**Presentation:** 3 good
**Contribution:** 3 good
**Rating:** 6
**Confidence:** 2

**Summary:**

This research delves into the realm of conditional stochastic optimization within the framework of federated learning. It begins by introducing the Federated Conditional Stochastic Gradient (FCSG) algorithm, which utilizes a conditional stochastic gradient estimator. Subsequently, the algorithm is refined by incorporating a momentum term into the update rules, thus successfully attenuating the influence of gradient estimate noise. Moreover, a variance reduction technique is validated to enhance sample complexity. The efficacy of the proposed algorithms is assessed through the evaluation of their performance on both real-world and synthetic datasets.

**Strengths:**

- The objective function mentioned in equation (1) appears to be a generic representation capable of encompassing a broad spectrum of stochastic optimization problems, including stochastic compositional stochastic optimization. Moreover, it lends itself well to implementations based on federated learning.

- The accelerated algorithm exhibits a sample complexity that aligns with the lower bound established for the corresponding single-machine counterpart.

- Simulated tests on the robust logistic regression and MAML tasks show the superior performance of the proposed approaches.

**Weaknesses:**

- The proof of convergence to the stationary points, as presented in the supplementary material, can sometimes be challenging to follow due to the lack of clarity regarding the variables with respect to which the gradients are computed. For example, in equation (6), it is unclear which variable or function each gradient corresponds to. Additionally, there appears to be a mismatch in the dimensionalities of the terms, as the multiplication involves two $d \times 1$ vectors (where $d$ represents the size of the vector $x$).

- The submission states that the MAML problem is an example of conditional stochastic optimization. It is better to discuss the related works about federated meta-learning to highlight the novelty of this paper.

- Compared with the FCSG algorithm, the FCSG-M and accelerated FCSG-M approaches require both the local model parameters and gradients to be transmitted to the central server for the aggregation, which may lead to higher communication overhead.

- For the experimental results on invariant logistic regression, the performance of the proposed accelerated FCSG-M method should be involved.

**Questions:**

See weaknesses.

**Limitations:**

certain limitations are discussed.

---

> ### Author Rebuttal · Authors · 2023-08-09
>
> We are deeply thankful for your appreciation and valuable comments.
>
> > Q1. "in equation (6), it is unclear which variable or function each gradient corresponds to ..."
>
> Thanks. I will fix these typos in the final version. Since our objective function is a CSO and its gradient follow the chain rule in the derivative of the composition of two differentiable functions.
>
> Given that Inner-layer function  $\mathbb{E}\_{\eta^n \mid \xi^n} g\_{\eta^n}^{n}(\cdot, \xi^n) : \mathbb{R}^{d} \rightarrow \mathbb{R}^{d^{\prime}}$ and the outer-layer function $\mathbb{E}\_{\xi^n} f\_{\xi^n}^{n}(\cdot): \mathbb{R}^{d^{\prime}} \rightarrow \mathbb{R}$, the dimension of $\nabla g$ is $d^{\prime} \times d$ and  dimension of $\nabla_g f$ is $d^{\prime} \times 1$. we rewrite (6) as
>
> $ \nabla F^n(x) = \nabla \mathbb{E}\_{\xi^n}\left[f^n\_{\xi^n}(g^n(x, \xi^n))\right] = \mathbb{E}\_{\xi^n}\left[\nabla\left(f^n\_{\xi^n}(g^n(x, \xi^n))\right)\right]$   = $\mathbb{E}\_{\xi^n}  [ [\nabla g^n(x, \xi^n) ]^{\top} \cdot \nabla_g f^n_{\xi^n}(g^n(x, \xi^n)) ]$
>
> >Q2. "It is better to discuss the related works about federated meta-learning to highlight the novelty of this paper."
>
> Thanks for your suggestion. We talked about some existing works [8]. Due to the page limitation,  we will add more related works in the final version.
>
> > Q3. " ...FCSG-M and accelerated FCSG-M approaches require both the local model parameters and gradients to be transmitted to .., which may lead to higher communication overhead"
>
> FCSG-M and accelerated FCSG-M need to send gradient estimators. But it is very common in existing FL algorithms [2][3][4]. I will add this in the limitation part.
>
> > Q4.   "For the experimental results on invariant logistic regression, the performance of the proposed accelerated FCSG-M method should be involved."
>
> The target of the first experiment is to evaluate the benefit of momentum and the effect of inner batch size. The FCSG-M method is relatively simple to explore. Since we cannot upload the external link to present images, I will add the results with the accelerated FCSG-M method in the final version.
>
> [1] On the convergence of local stochastic compositional gradient descent with momentum. In International Conference on Machine Learning, pages 7017–7035. PMLR,327 2022
>
> [2] Khanduri, Prashant, et al. "Stem: A stochastic two-sided momentum algorithm achieving near-optimal sample and communication complexities for federated learning." Advances in Neural Information Processing Systems 34 (2021): 6050-6061.
>
> [3] Karimireddy, Sai Praneeth, et al. "Scaffold: Stochastic controlled averaging for federated learning." International Conference on Machine Learning. PMLR, 2020.
>
> [4] Sharma, P., Panda, R., Joshi, G. and Varshney, P., 2022, June. Federated minimax optimization: Improved convergence analyses and algorithms. In International Conference on Machine Learning. PMLR.

---

> > ### Comment · Reviewer_VagV · 2023-08-16
> > **Response to author comments**
> >
> > Thank you for your answers.
> >
> > I've gone through your responses and would like to provide the following feedback:
> > 1. I am pleased to see that you addressed the ambiguities in equation (6). The chain rule clarification, coupled with the adding the missing transposition, and discussion regarding the dimensions of "f" and "g" offers clearer insights. The updated representation of the equation undeniably enhances clarity. Additionally, similar concerns might arise later in the proof, so it's advisable to review the proof and adjust the notation where necessary.
> >
> > 2. I appreciate your willingness to add more related works in the final version.
> >
> > 3. It is reassuring to know that the practice of sending gradient estimators, as seen in the FCSG-M and accelerated FCSG-M algorithms, aligns with other Federated Learning algorithms. Including this in the limitations section, can be valuable for readers.
> >
> > 4. The clarification regarding the focus of the first experiment is appreciated. I look forward to seeing the results from the accelerated FCSG-M method in the final version. It would provide a more rounded view of the methodologies you've explored.
> >
> > I think the paper presents a noteworthy contribution and is generally a sound paper. Reiterating the observation of reviewers jBXN14 and pkoz, the inclusion of additional example problems in the introduction would enhance the motivation behind CSO. While I am confident in my current score, I remain open to adjusting it upwards should further motivation be provided.

---

> > > ### Author Response · Authors · 2023-08-17
> > > **Thanks for your reponse**
> > >
> > > Thank you very much for your reply.
> > >
> > > > Q1: " ... similar concerns might arise later in the proof, so it's advisable to review the proof and adjust the notation where necessary..."
> > >
> > > Thanks for your suggestion. We do check other parts in the proof and fix related typos.
> > >
> > > > Q " ... the inclusion of additional example problems in the introduction would enhance the motivation behind CSO. ..."
> > >
> > > Thanks. Our extra experiments about online  AUPRC maximization in FL are added below. Existing AUPRC algorithms maintain an inner state for each specific data point. In this case, these states are different in different clients and also cost much memory. For the large-scale distributed data over multiple clients, algorithms for online AUPRC maximization in FL is necessary.
> > >
> > > Following the eq (2) in [1], the online AUPRC maximization could be reformulated as:
> > >
> > > $ \begin{align}
> > > \min P = - \mathbb{E}\_{\mathbf{x_i} \sim  \mathcal{D}_{+}} \frac{\mathbb{E}\_{\mathbf{x_s} \sim  \mathcal{D}} \mathbf{I}\left(y_s=1\right)  \ell \left(\mathbf{w}; \mathbf{x}_s, \mathbf{x}_i \right)}{\mathbb{E}\_{\mathbf{x_s} \sim \mathcal{D}} \quad \ell\left(\mathbf{w}; \mathbf{x}_s, \mathbf{x}_i\right)}
> > > \end{align}$
> > >
> > > Following the eq(4) in [1], FL objective function is listed below:
> > > $\min \frac{1}{K} \sum P\_k (\mathbf{w}) = \min \frac{1}{K} \sum\mathbb{E}\_{\mathbf{x_i} \sim  \mathcal{D}^k\_{+}} [f^k(g^k_{\mathbf{x}_i} (\mathbf{w}))]$
> > >
> > > It is a two-level problem and $x_i$ is sampled from positive datasets and $x_s$ is sampled to estimate the score rank of $x_i$ . The inner objective depends on both $x_i$ and $x_s$. Since federated AUPRC is a special example of a federated CSO, our algorithms could be directly applied to it.
> > >
> > > As AUROC maximization in federated settings has been demonstrated in existing works, e.g., [2, 3]. we use CODA+ in [3] as a baseline. In addition, we also use FedAvg with cross-entropy loss as a baseline. Since we cannot submit the external link, we will release related code in the final version.
> > >
> > > We choose MNIST dataset and CIFAR-10 dataset. Given that we cannot upload the figures, we make a table to show our results.
> > > Since AUPRC is used for binary classification, the first half of the classes in the MNIST and CIFAR10 datasets are designated to be a negative class, and the rest half of the classes are considered to be the positive class. Then, we remove 80% of the positive examples in the training set to make it imbalanced, while keeping the test set unchanged. The results show that our algorithms could be used to solve the online AUPRC maximization in FL and it largely improves the model's performance. In the final version, we will also add the test AP vs training iteration image to show the convergence rate for different algorithms.
> > >
> > > | Datasets |  SGD + CE |  CODA+ | FCSG | FCSG-M | Acc-FCSG-M |
> > > | ---------- | ------------- | --------- | ------- | ---------- | ---------------- |
> > > | MNIST    |     0.9357    | 0.9733   |0.9868 | 0.9878     |   0.9879           |
> > > | CIFAR-10| 0.5059       | 0.6039  |  0.7130 |  0.7157    |  0.7184           |
> > >
> > >
> > > [1] Qi, Q., Luo, Y., Xu, Z., Ji, S., and Yang, T. Stochastic optimization of areas under precision-recall curves with provable convergence. Advances in Neural Information Processing Systems, 2021. https://arxiv.org/pdf/2104.08736.pdf
> > >
> > > [2] Guo, Zhishuai, Mingrui Liu, Zhuoning Yuan, Li Shen, Wei Liu, and Tianbao Yang. "Communication-efficient distributed stochastic auc maximization with deep neural networks." In International conference on machine learning, pp. 3864-3874. PMLR, 2020.
> > >
> > > [3] Z Yuan, Z Guo, Y Xu, Y Ying, T Yang. Federated Deep AUC Maximization with a Constant Communication Complexity, In International conference on machine learning, 2021.

---

> > > > ### Comment · Reviewer_VagV · 2023-08-19
> > > > **Thanks for your response**
> > > >
> > > > Thank you for the additional experiments on online AUPRC maximization in FL. The outcomes are encouraging and could be used to enhance the overall motivation. I've revised my rating accordingly.

---

> > > > > ### Author Response · Authors · 2023-08-19
> > > > >
> > > > > We are deeply thankful for your constructive review.

---

### Official Review · Reviewer_L363 · 2023-07-12

**Soundness:** 4 excellent
**Presentation:** 3 good
**Contribution:** 3 good
**Rating:** 7
**Confidence:** 4

**Summary:**

Authors propose methods for problem of stochastic optimisation of composition of functions' expected values, which can be addressed with conditional gradient concept. Federated algorithm, which uses biased gradient estimation, and its modifications with momentum and acceleration are proposed. Theoretical guarantees on iteration and oracle complexities match lower bounds or correspond to best-known results.

**Strengths:**

The algorithms proposed are efficient theoretically and their efficiency is confirmed by numerical experiments. The experiments show practically interesting problems statements.

**Weaknesses:**

Empirical study could be more extensive nd helpful for practitioners who may want to use your method in their tasks if the comparison contained also different approaches designed specifically for MAML.

**Questions:**

It may be better if convergence curves of algorithms were provided with shadows showing deviation of trajectory for different runs with different realisations of randomness.

**Limitations:**

Everything is okay.

---

> ### Author Rebuttal · Authors · 2023-08-08
>
> We are deeply thankful for your comments. I will add deviation of trajectory for different runs with different realisations of randomness in our final version.

---

> > ### Comment · Reviewer_L363 · 2023-08-19
> >
> > Dear authors, thank you for your work on the final version of your paper! The rebuttal has clarified my questions. I decided to keep my overall rating the same.

---

> > > ### Author Response · Authors · 2023-08-19
> > >
> > > Thanks a lot for your appreciation.

---

### Decision · Program_Chairs · 2023-09-21

**Decision:**

Accept (poster)

**Comment:**

Dear authors,

Thank you for submitting your paper focused on federated conditional stochastic optimization. The reviewers appreciated many of your new contributions, including combining the idea from STORM and extending it into a federated setting with a conditional stochastic flavor. Despite this, the reviewers also pointed out several weaknesses in your work, many of which you successfully addressed during the rebuttal phase. Some of the reviewers increased their scores and agreed with acceptance only based on the promise that the final camera-ready version would include the amendments you promised and demonstrated. I would also encourage you to highlight your contributions in the final camera-ready version to demonstrate further that your paper is not just a trivial extension of STORM+CSO for FL.

Best,
AC